# Random Shuffling Beats SGD Only After Many Epochs on Ill-Conditioned Problems

**Itay Safran**
Princeton University
isafran@princeton.edu

**Ohad Shamir**
Weizmann Institute of Science
ohad.shamir@weizmann.ac.il

## Abstract

Recently, there has been much interest in studying the convergence rates of without-replacement SGD, and proving that it is faster than with-replacement SGD in the worst case. However, known lower bounds ignore the problem's geometry, including its condition number, whereas the upper bounds explicitly depend on it. Perhaps surprisingly, we prove that when the condition number is taken into account, without-replacement SGD does not significantly improve on with-replacement SGD in terms of worst-case bounds, unless the number of epochs (passes over the data) is larger than the condition number. Since many problems in machine learning and other areas are both ill-conditioned and involve large datasets, this indicates that without-replacement does not necessarily improve over with-replacement sampling for realistic iteration budgets. We show this by providing new lower and upper bounds which are tight (up to log factors), for quadratic problems with commuting quadratic terms, precisely quantifying the dependence on the problem parameters.

## 1 Introduction

We consider solving finite-sum optimization problems of the form

$$F(\mathbf{x}) \;=\; \frac{1}{n}\sum_{i=1}^{n} f_i(\mathbf{x})$$

using stochastic gradient descent (SGD). Such problems are extremely common in modern machine learning (e.g., for empirical risk minimization), and stochastic gradient methods are the most popular approach for large-scale problems, where both $n$ and the dimension are large. The classical approach to apply SGD to such problems is to repeatedly sample indices $i \in \{1, \ldots, n\}$ uniformly at random, and perform updates of the form $\mathbf{x}_{t+1} = \mathbf{x}_t - \eta_t \nabla f_i(\mathbf{x}_t)$, where $\eta_t$ is a step-size parameter. With this sampling scheme, each $\nabla f_i(\mathbf{x}_t)$ is a random unbiased estimate of $\nabla F(\mathbf{x}_t)$ (conditioned on $\mathbf{x}_t$). Thus, the algorithm can be seen as a "cheap" noisy version of plain gradient descent on $F(\cdot)$, where each iteration requires computing the gradient of just a single function $f_i(\cdot)$, rather than the gradient of $F(\cdot)$ (which would require computing and averaging $n$ individual gradients). This key observation facilitates the analysis of SGD, while simultaneously explaining why SGD is much more efficient than gradient descent on large-scale problems.

However, the practice of SGD differs somewhat from this idealized description: In practice, it is much more common to perform without-replacement sampling of the indices, by randomly shuffling the indices $\{1, \ldots, n\}$ and processing the individual functions in that order – that is, choosing a random permutation $\sigma$ on $\{1, \ldots, n\}$, and performing updates of the form $\mathbf{x}_{t+1} = \mathbf{x}_t - \eta_t \nabla f_{\sigma(t)}(\mathbf{x}_t)$. After a full pass over the $n$ functions, further passes are made either with the same permutation $\sigma$ (known as single shuffling), or with a new random permutation $\sigma$ chosen before each pass (known as random

Table 1: Upper bounds on the expected optimization error for quadratic strongly-convex problems, ignoring constants and log factors. The upper table corresponds to random reshuffling, and the lower table corresponds to single shuffling. The right column presents a necessary (possibly non-sufficient) condition for the bound to be valid and smaller than that of with-replacement SGD, which is order of $1/(\lambda nk)$ [Shamir and Zhang, 2013, Jain et al., 2019a]. An asterisk (*) denotes results where the bound is on the expected squared distance to the global minimum, rather than optimization error. For squared distance, the corresponding bound for with-replacement SGD is order of $1/(\lambda^2 nk)$ [Nemirovski et al., 2009, Rakhlin et al., 2012].

| Paper | Bound | Improves on with-Replacement? |
|---|---|---|
| Gürbüzbalaban et al. [2015b](*) | $\frac{1}{(\lambda k)^2}$ | Only if $k \gtrsim n$ |
| HaoChen and Sra [2018] | $\frac{1}{\lambda^4}\left(\frac{1}{(nk)^2} + \frac{1}{k^3}\right)$ if $k \gtrsim \frac{1}{\lambda}$ | Only if $k \gtrsim \frac{1}{\lambda} \cdot \max\{1, \sqrt{\frac{n}{\lambda}}\}$ |
| Rajput et al. [2020] | $\frac{1}{\lambda^4}\left(\frac{1}{(nk)^2} + \frac{1}{nk^3}\right)$ if $k \gtrsim \frac{1}{\lambda^2}$ | Only if $k \gtrsim \frac{1}{\lambda^2}$ |
| Ahn et al. [2020] | $\frac{1}{\lambda^4}\left(\frac{1}{(nk)^2} + \frac{1}{\lambda^2 nk^3}\right)$ | Only if $k \gtrsim \frac{1}{\lambda^{2.5}}$ |

| Paper | Bound | Improves on with-Replacement? |
|---|---|---|
| Gürbüzbalaban et al. [2015a](*) | $\frac{1}{(\lambda k)^2}$ | Only if $k \gtrsim n$ |
| Ahn et al. [2020] | $\frac{1}{\lambda^4 nk^2}$ if $k \gtrsim \frac{1}{\lambda^2}$ | Only If $k \gtrsim \frac{1}{\lambda^2}$ |
| Mishchenko et al. [2020](*) | $\frac{1}{\lambda^3 nk^2}$ | Only if $k \gtrsim \frac{1}{\lambda}$ |

reshuffling). Such without-replacement schemes are not only more convenient to implement in many cases, they often also exhibit faster error decay than with-replacement SGD [Bottou, 2009, Recht and Ré, 2012]. Unfortunately, analyzing this phenomenon has proven to be notoriously difficult. This is because without-replacement sampling creates statistical dependencies between the iterations, so the stochastic gradients computed at each iteration can no longer be seen as unbiased estimates of gradients of $F(\cdot)$.

Nevertheless, in the past few years our understanding of this problem has significantly improved. For concreteness, let us focus on a classical setting where each $f_i(\cdot)$ is a convex quadratic, and $F(\cdot)$ is strongly convex. Suppose that SGD is allowed to perform $k$ epochs, each of which consists of passing over the $n$ individual functions in a random order. Assuming the step size is appropriately chosen, it has been established that SGD with single-shuffling returns a point whose expected optimization error is on the order of $1/(nk^2)$. For random reshuffling, this further improves to $1/(nk)^2 + 1/(nk^3)$ (see further discussion in the related work section below). In contrast, the expected optimization error of with-replacement SGD after the same overall number of iterations ($nk$) is well-known to be on the order of $1/(nk)$ (e.g., Nemirovski et al. [2009], Rakhlin et al. [2012]). Moreover, these bounds are known to be unimprovable in general, due to the existence of nearly matching lower bounds [Safran and Shamir, 2020, Rajput et al., 2020]. Thus, as the number of epochs $k$ increases, without-replacement SGD seems to provably beat with-replacement SGD.

Despite these encouraging results, it is important to note that the bounds as stated above quantify only the dependence on $n, k$, and ignore dependencies on other problem parameters. In particular, it is well-known that the convergence of SGD is highly sensitive to the problem's strong convexity and smoothness parameters, their ratio (a.k.a. the condition number), as well as the magnitude of the gradients. Unfortunately, existing lower bounds ignore some of these parameters (treating them as constants), while in known upper bounds these can lead to vacuous results in realistic regimes. To give a concrete example, let us return to the case where each $f_i(\cdot)$ is a convex quadratic function, and assume that $F(\cdot)$ is $\lambda$-strongly convex for some $\lambda > 0$. In Table 1, we present existing upper bounds for this case, as a function of both $n, k$ as well as $\lambda$ (fixing other problem parameters, such as the

smoothness parameter, as constants).[1] The important thing to note about these bounds is that they are valid and improve over the bound for with-replacement SGD only when the number of epochs $k$ is at least $n$ or $1/\lambda$ (or even more). Unfortunately, such a regime is problematic for two reasons: First, large-scale high-dimensional problems are often ill-conditioned, with $\lambda$ being very small and $n$ being very large, so requiring that many passes $k$ over all functions can easily be prohibitively expensive. Second, if we allow $k > \frac{1}{\lambda}$, there exist much better and simpler methods than SGD: Indeed, we can simply run deterministic gradient descent for $k$ iterations (computing the gradient of $F(\cdot)$ at each iteration by computing and averaging the gradients of $f_i(\cdot)$ in any order we please). Since the optimization error of gradient descent (as a function of $k, \lambda$) scales as $\exp(-\lambda k)$ [Nesterov, 2018], we compute a nearly exact optimum (up to error $\epsilon$) as long as $k \gtrsim \frac{1}{\lambda} \log(\frac{1}{\epsilon})$. Moreover, slightly more sophisticated methods such as accelerated gradient descent or the conjugate gradient method enjoy an error bound scaling as $\exp(-k\sqrt{\lambda})$, so in fact, we can compute an $\epsilon$-optimal point already when $k \gtrsim \frac{1}{\sqrt{\lambda}} \log(\frac{1}{\epsilon})$.

Thus, the power of SGD is mostly when $k$ is relatively small, and definitely smaller than quantities such as $1/\lambda$. However, the upper bounds discussed earlier do not imply any advantage of without-replacement sampling in this regime. Of course, these are only upper bounds, which might possibly be loose. Thus, it is not clear if this issue is simply an artifact of the existing analyses, or a true issue of without-replacement sampling methods.

In this paper, we rigorously study this question, with the following (perhaps surprising) conclusion: At least in the worst-case, without-replacement schemes do not significantly improve over with-replacement sampling, unless the number of epochs $k$ is larger than the problem's condition number (which suitably defined, scales as $1/\lambda$). As discussed above, this implies that the expected benefit of without-replacement schemes may not be manifest for realistic iteration budgets. In more detail, our contributions are as follows:

- We prove that there is a simple quadratic function $F(\cdot)$, which is $\lambda$-strongly convex and $\lambda_{\max}$-smooth, such that the expected optimization error of SGD with single shuffling (using any fixed step size) is at least

$$\Omega\left(\frac{1}{\lambda n k} \cdot \min\left\{1, \frac{\lambda_{\max}/\lambda}{k}\right\}\right) .$$

  (See Thm. 1.) Comparing this to the $\Theta(1/(\lambda n k))$ bound for with-replacement SGD, we see that we cannot possibly get a significant improvement unless $k$ is larger than the condition number term $\frac{\lambda_{\max}}{\lambda}$.

- For SGD with random reshuffling, we prove a similar lower bound of

$$\Omega\left(\frac{1}{\lambda n k} \cdot \min\left\{1, \frac{\lambda_{\max}/\lambda}{n k} + \frac{\lambda_{\max}^2/\lambda^2}{k^2}\right\}\right) .$$

  (See Thm. 2.) As before, this improves on the $\Theta(1/(\lambda n k))$ bound of with-replacement SGD only when $k > \frac{\lambda_{\max}}{\lambda}$.

- We provide matching upper bounds in all relevant parameters (up to constants and log factors and assuming the input dimension is fixed – See Sec. 4), which apply to the class of quadratic functions with commuting quadratic terms (which include in particular the lower bound constructions above).

- To illustrate our theoretical results, we perform a few simple experiments comparing with- and without-replacement sampling schemes on our lower bound constructions (see Sec. 5). Our results accord with our theoretical findings, and show that if the number of epochs is not greater than the condition number, then without-replacement does not necessarily improve upon with-replacement sampling. Moreover, it is observed that for some of the values of $k$ exceeding the condition number by as much as $50\%$, with-replacement still provides comparable results to without-replacement, even when averaging their performance over many instantiations.

---

[1]Focusing only on $\lambda$ is enough for the purpose of our discussion here, and moreover, the dependence on other problem parameters is not always explicitly given in the results of previous papers.

We conclude with a discussion of the results and open questions in Sec. 6.

We note that our lower and upper bounds apply to SGD using a fixed step size, $\eta_t = \eta$ for all $t$, and for the iterate reached after $k$ epochs. Thus, they do not exclude the possibility that better upper bounds can be obtained with a variable step size strategy, or some iterate averaging scheme. However, we conjecture that it is not true, as existing upper bounds either do not make such assumptions or do not beat the lower bounds presented here. Moreover, SGD with variable step sizes can generally be matched by SGD employing a fixed optimal step size (dependent on the problem parameters and the overall number of iterations).

### Related Work

Proving that without-replacement SGD converges faster than with-replacement SGD has been the focus of a line of recent works, which we briefly survey below (focusing for concreteness on quadratic and strongly convex problems, as we do in this paper). Gürbüzbalaban et al. [2015a,b] proved that without-replacement beats with-replacement in terms of the dependence on $k$ ($1/k^2$ vs. $1/k$, without specifying a decay in terms of $n$). Shamir [2016] proved that without-replacement is not worse than with-replacement in terms of dependence on $n$, but only for $k = 1$. HaoChen and Sra [2018] managed to prove a better bound for random reshuffling, scaling as $1/(nk)^2 + 1/k^3$. Safran and Shamir [2020] proved a lower bound of $1/(nk)^2 + 1/nk^3$ for random reshuffling and $1/(nk^2)$ for single shuffling, and also proved matching upper bounds in the (very) special case of one-dimensional quadratic functions. A series of recent works [Jain et al., 2019b, Rajput et al., 2020, Ahn et al., 2020, Mishchenko et al., 2020, Nguyen et al., 2020] showed that these are indeed the optimal bounds in terms of $n, k$, by proving matching upper bounds which apply to general quadratic functions and beyond. Rajput et al. [2020] were also able to show that for strongly convex and smooth functions beyond quadratics, the error rate provably becomes order of $1/(nk^2)$ for both single shuffling and random reshuffling.

A related and ongoing line of work considers the question of whether without-replacement SGD can be shown to be superior to with-replacement SGD, individually on any given problem of a certain type [Recht and Ré, 2012, Lai and Lim, 2020, De Sa, 2020, Yun et al., 2021]. However, this is different than comparing worst-case behavior over problem classes, which is our focus here. Without-replacement SGD was also studied under somewhat different settings than ours, such as Ying et al. [2018], Tran et al. [2020], Huang et al. [2021].

In our paper, we focus on the well-known and widely popular SGD algorithm, using various sampling schemes. However, we note that for finite-sum problems, different and sometimes better convergence guarantees can be obtained using other stochastic algorithms, such as variance-reduced methods, adaptive gradient schemes, or by incorporating momentum. Analyzing the effects of without-replacement sampling on such methods is an interesting topic (studied for example in Tran et al. [2020]), which is however outside the scope of our paper.

## 2   Preliminaries

**Notation and terminology.** We let bold-face letters (such as $\mathbf{x}$) denote vectors. For a vector $\mathbf{x}$, $x_j$ denotes its $j$-th coordinate. For natural $n$, let $[n]$ be shorthand for the set $\{1, \ldots, n\}$. For some vector $\mathbf{v}$, $\|\mathbf{v}\|$ denotes its Euclidean norm. We let $\|\cdot\|_{\text{sp}}$ denote the spectral norm of a matrix. A twice-differentiable function $f$ on $\mathbb{R}^d$ is $\lambda$-strongly convex, if its Hessian satisfies $\nabla^2 F(\mathbf{x}) \succeq \lambda I$ for all $\mathbf{x}$, and is $L$-smooth if its gradient is $L$-Lipschitz. $f$ is quadratic if it is of the form $f(\mathbf{x}) = \frac{1}{2}\mathbf{x}^\top A \mathbf{x} + \mathbf{b}^\top \mathbf{x} + c$ for some matrix $A$, vector $\mathbf{b}$ and scalar $c$.[2] Note that if $A$ is a PSD matrix, then the strong convexity and smoothness parameters of $f$ correspond to the smallest and largest eigenvalues of $A$, respectively. Moreover, the ratio between the largest and smallest eigenvalues is known as the condition number of $f$. We use standard asymptotic notation $\Theta(\cdot), \mathcal{O}(\cdot), \Omega(\cdot)$ to hide constants, and $\tilde{\Theta}(\cdot), \tilde{\mathcal{O}}(\cdot), \tilde{\Omega}(\cdot)$ to hide constants and logarithmic factors.

**SGD.** As mentioned in the introduction, we focus on plain SGD using a constant step size $\eta$, which performs $k$ epochs, and in each one takes $n$ stochastic gradients steps w.r.t. the functions

---

[2]We note that throughout the paper we omit the constant scalar terms, since these do not affect optimization when performing SGD (only the optimal value attained).

$f_{\sigma(1)}, \ldots, f_{\sigma(n)}$, with $\sigma$ being a random permutation which is either sampled afresh at each epoch (random reshuffling) or chosen once and then used for all $k$ epochs (single shuffling). We let $\mathbf{x}_0$ denote the initialization point, and $\mathbf{x}_1, \ldots, \mathbf{x}_k$ denote the iterates arrived at the end of epoch $1, \ldots, k$ respectively.

## 3 Lower Bounds

In this section, we formally present our lower bounds for single shuffling and random reshuffling SGD. Our lower bounds will use a particularly simple class of quadratic functions, which satisfy the following assumptions.

**Assumption 1** (Lower Bounds Assumption). *$F(\mathbf{x})$ is $\lambda$-strongly convex and of the form $\frac{1}{n}\sum_{i=1}^n f_i(\mathbf{x})$, and each $f_i$ is of the form $f_i(\mathbf{x}) := \sum_j \frac{1}{2}a_j x_j^2 - b_j x_j$, where for all $j$, $a_j \in [\lambda, \lambda_{\max}] \cup \{0\}$ and $b_j \in [-\frac{G}{2}, \frac{G}{2}]$. Suppose $nk$ is large enough so that $\frac{\log(nk)L}{\lambda nk} \leq 1$. We assume that the algorithm is initialized at some $\mathbf{x}_0$ on which $\|\nabla F(\mathbf{x}_0)\| \leq G$.*

Note that any such function $F$ is $\lambda_{\max}$-smooth, and so are its components. We emphasize that since our lower bounds apply to such functions, they also automatically apply to larger function classes (e.g. more general quadratic functions, the class of all $\lambda$-strongly convex and $\lambda_{\max}$-smooth functions, etc). Moreover, our lower bounds also apply in harder and more general settings where for example only a partial or noisy view of the Hessian of the functions is revealed.

Having stated our assumptions, we now turn to present our results for this section, beginning with our single shuffling lower bound.

**Theorem 1** (Single Shuffling Lower Bound). *For any $k \geq 1, n > 1$, and positive $G, \lambda$, there exist a function $F$ on $\mathbb{R}^2$ and an initialization point $\mathbf{x}_0$ satisfying Assumption 1, for which single shuffling SGD using any fixed step size $\eta > 0$ satisfies*

$$\mathbb{E}\left[F(\mathbf{x}_k)\right] \geq c\frac{G^2}{\lambda nk}\min\left\{1, \frac{\lambda_{\max}/\lambda}{k}\right\},$$

*for some universal constant $c > 0$.*

The function $F$ used in the above theorem is given by the component functions

$$f_i(\mathbf{x}) = f_i(x_1, x_2) := \frac{\lambda}{2}x_1^2 + \frac{\lambda_{\max}}{2}x_2^2 + \begin{cases} \frac{G}{2}x_2 & i \leq \frac{n}{2} \\ -\frac{G}{2}x_2 & i > \frac{n}{2} \end{cases}. \tag{1}$$

For the random reshuffling sampling scheme, we have the following theorem.

**Theorem 2** (Random Reshuffling Lower Bound). *For any $k \geq 1, n > 1$, and positive $G, \lambda$, there exist a function $F$ on $\mathbb{R}^3$ and an initialization point $\mathbf{x}_0$ satisfying Assumption 1, for which random reshuffling SGD using any fixed step size $\eta > 0$ satisfies*

$$\mathbb{E}\left[F(\mathbf{x}_k)\right] \geq c\frac{G^2}{\lambda nk}\min\left\{1, \frac{\lambda_{\max}/\lambda}{nk} + \frac{\lambda_{\max}^2/\lambda^2}{k^2}\right\},$$

*for some universal constant $c > 0$.*

The function $F$ used in the above theorem is given by the component functions

$$f_i(\mathbf{x}) = f_i(x_1, x_2, x_3) := \frac{\lambda}{2}x_1^2 + \frac{\lambda_{\max}}{2}x_2^2 + \begin{cases} \frac{G}{2}x_2 + \frac{\lambda_{\max}}{2}x_3^2 + \frac{G}{2}x_3 & i \leq \frac{n}{2} \\ -\frac{G}{2}x_2 - \frac{G}{2}x_3 & i > \frac{n}{2} \end{cases}. \tag{2}$$

In contrast, with-replacement SGD bounds are of the form $\tilde{\mathcal{O}}(G^2/(\lambda nk))$.[3] Thus, we see that for both single shuffling and random shuffling, we cannot improve on the worst-case performance of

---

[3]There is some subtlety here, as most such bounds are either on some average of the iterates rather than the last iterate, or use a non-constant step-size (in which case it is possible to prove a bound of $\mathcal{O}(G^2/\lambda nk)$). However, Shamir and Zhang [2013] prove a bound of $\tilde{\mathcal{O}}(G^2/\lambda nk)$ (involving an additional $\log(nk)$ factor) for the last iterate, using a variable step-size $\eta_t = 1/(\lambda t)$, and it is not difficult to adapt the analysis to SGD with a constant step size $\eta = \tilde{\Theta}(1/(\lambda nk))$.

with-replacement SGD by more than constants or logarithmic terms, unless $k$ is larger than the condition number parameter $\frac{\lambda_{\max}}{\lambda}$. As discussed earlier, this can be an unrealistically large regime for $k$ in many cases.

The proofs of the above theorems, which appear in Appendices A.1 and A.2, follow a broadly similar strategy to the one-dimensional constructions in Safran and Shamir [2020], with the crucial difference that we construct a multivariate function having different curvature in each dimension. The constructions themselves are very simple, but the analysis is somewhat technical and intricate. Roughly speaking, our analysis splits into three different cases, where in each a different magnitude of the step size is considered. In the first case, where the step size is too small, we use a dimension with small curvature to show that if we initialize far enough from the global minimizer then SGD will converge too slowly. The construction in the remaining dimensions is similar to that in Safran and Shamir [2020], and uses a larger curvature than the first dimension to generate dependency on the condition number, due to the somewhat too large step size that is dictated by the first case. In the remaining two cases, the step size is either too large and the resulting bound is not better than with-replacement SGD; or is chosen in a manner which achieves better dependence on $n, k$, but also incurs additional error terms dependent on $\lambda_{\max}, \lambda$, due to the statistical dependencies between the gradients created via without-replacement sampling. Combining the cases leads to the bounds stated in the theorems.

We also make the following remarks regarding our lower bounds:

**Remark 1** (Separating $\lambda_{\max}$ and $L$). *Our parameters satisfy the following chain of inequalities $\lambda \leq \lambda_{\max} \leq L$. Since the condition number is commonly defined in the literature as $L/\lambda$, we note that in our lower bound constructions we have $\lambda_{\max} = L$. However, since $\lambda_{\max}$ could potentially be smaller than $L$, this raises the question of whether it is possible to construct a lower bound in which $\lambda_{\max}$ is sufficiently smaller than $L$ yet the lower bound depends on $L$. The upper bounds we present in the next section indicate that this is not possible when the $A_i$'s commute, since in this case we get bounds with no dependence on $L$ (at least when $nk$ is large enough). This implies that to separate the two quantities $\lambda_{\max}, L$, one would necessarily need a construction where the $A_i$'s do not commute. We leave the derivation of such a construction to future work.*

**Remark 2** (Bound on $L$). *We note that the condition $\frac{\log(nk)L}{\lambda nk} \leq 1$ is equivalent to requiring $nk$ to be at least on the order of the condition number $L/\lambda$ (up to log factors). This is a mild requirement, since if $nk$ is smaller than the condition number, then even if $f_i = f$ for all $i$ (that is, we perform deterministic gradient descent on the function $f$), there are quadratic functions for which no non-trivial guarantee can be obtained [Nesterov, 2018].*

**Remark 3** (A Random Reshuffling Lower Bound in $\mathbb{R}^2$). *Our lower bound construction for single shuffling is a function in $\mathbb{R}^2$, whereas our random reshuffling construction is in $\mathbb{R}^3$. We believe that a construction in $\mathbb{R}^2$ is also possible for random reshuffling, however this would require a more technically involved proof which for example also lower bounds the right-most summand in Eq. (5).*

## 4 Matching Upper Bounds

In this section, we provide upper bounds that match our lower bounds from the previous section (up to constants and log factors, and for a fixed input dimension). Whereas our lower bounds use quadratic constructions where each matrix $A_i$ is diagonal, here we prove matching upper bounds in the somewhat more general case where the quadratic terms commute. Before we state our upper bounds, however, we will first state and discuss the assumptions used in their derivation.

**Assumption 2** (Upper Bounds Assumption). *Suppose the input dimension $d$ is a fixed constant. Assume $F(\mathbf{x}) = \frac{1}{2}\mathbf{x}^\top A\mathbf{x} - \mathbf{b}^\top \mathbf{x}$ is a quadratic finite-sum function of the form $\frac{1}{n}\sum_{i=1}^n f_i(\mathbf{x})$ for some $n > 1$, which is $\lambda$-strongly convex and satisfies $\|A\|_{\mathrm{sp}} = \lambda_{\max}$. Each $f_i$ is a convex quadratic function of the form $f_i(x) = \frac{1}{2}\mathbf{x}^\top A_i\mathbf{x} - \mathbf{b}_i^\top \mathbf{x}$, for commuting, symmetric, and PSD matrices $A_i \in \mathbb{R}^{d \times d}$, which all satisfy $\|A_i\|_{\mathrm{sp}} \leq L$. Moreover, suppose that $\|\nabla f_i(\mathbf{x}^*)\| \leq G$ for all $i \in [n]$, where $\mathbf{x}^* := \arg\min_\mathbf{x} F(\mathbf{x})$ is the global minimizer. Assume $nk$ is large enough so that $\frac{\log(nk)L}{\lambda nk} \leq 1$. We assume that the algorithm is initialized at some $\mathbf{x}_0$ on which $\|\nabla F(\mathbf{x}_0)\| \leq G$.*

Note that our lower bound constructions satisfy the assumptions above. We remark that as far as upper bounds go, these assumptions are somewhat different than those often seen in the literature. E.g. we require that $\nabla f_i$ is bounded only at the global minimizer $\mathbf{x}^*$, rather than in some larger domain as

commonly assumed in the literature. However this comes at a cost of assuming that $d$ is fixed. We remark that we believe that it is possible to extend our upper bounds to hold for arbitrary $d$ with no dependence on $d$, but this would require a more technically involved analysis (e.g. a multivariate version of Proposition 3). Nevertheless, the bounds presented here are sufficient for the purpose of establishing the tightness of our lower bound constructions, which involve 2-3 dimensions. Lastly, we recall that the assumption on the magnitude of $nk$ is very mild as explained in Remark 2.

We now present the upper bound for single shuffling SGD:

**Theorem 3** (Single Shuffling Upper Bound). *Suppose $F(\mathbf{x}) = \frac{1}{n} \sum_{i=1}^{n} f_i(\mathbf{x})$ satisfy Assumption 2, and fix the step size $\eta = \frac{\log(nk)}{\lambda nk}$. Then single shuffling SGD satisfies*

$$\mathbb{E}\left[F(\mathbf{x}_k)\right] \;\leq\; \tilde{\mathcal{O}}\left(\frac{G^2}{\lambda nk} \cdot \min\left\{1\,,\,\frac{\lambda_{\max}/\lambda}{k}\right\}\right)\,,$$

*where the $\tilde{\mathcal{O}}$ hides a universal constant, factors logarithmic in $n, k, \lambda_{\max}, 1/\lambda$, and a factor linear in $d$.*

Next, we have the following result for random reshuffling SGD.

**Theorem 4** (Random Reshuffling Upper Bound). *Suppose $F(\mathbf{x}) = \frac{1}{n} \sum_{i=1}^{n} f_i(\mathbf{x})$ satisfy Assumption 2, and fix the step size $\eta = \frac{\log(nk)}{\lambda nk}$. Then random reshuffling SGD satisfies*

$$\mathbb{E}\left[F(\mathbf{x}_k)\right] \;\leq\; \tilde{\mathcal{O}}\left(\frac{G^2}{\lambda nk} \cdot \min\left\{1\,,\,\frac{\lambda_{\max}/\lambda}{nk} + \frac{\lambda_{\max}^2/\lambda^2}{k^2}\right\}\right)\,,$$

*where the $\tilde{\mathcal{O}}$ hides a universal constant, factors logarithmic in $n, k, \lambda_{\max}, 1/\lambda$, and a factor linear in $d$.*

The proofs of the above theorems, which appear in Appendices A.3 and A.4, are based on reducing our problem to a one-dimensional setting, deriving a closed-form expression for $x_k$ (as was done in Safran and Shamir [2020]), and then carefully bounding each term in the resulting expression. However, our bounds refine those of Safran and Shamir [2020], allowing us to achieve upper bounds that precisely capture the dependence on all the problem parameters. Both proofs split the analysis into two cases: (i) The case where we perform sufficiently many epochs ($k > \lambda_{\max}/\lambda$) and are able to improve upon with-replacement SGD; and (ii), where too few epochs are performed ($k < \lambda_{\max}/\lambda$), in which case the sub-optimality rate matches that of with-replacement SGD. In both theorems, new tools to control the random quantities that arise during the problem analysis are required to attain bounds that match our lower bounds (see Proposition 3 in the Appendix for example). We point out that the choice of step size in the theorems is not unique and that the same upper bounds, up to log factors, apply to other choices of the step size (we did not attempt to optimize the logarithmic terms in the bounds).

It is interesting to note that somewhat surprisingly, our upper bounds do not depend on $L$, only on the smaller quantity $\lambda_{\max}$. $L$ only appears via the condition $\eta L \leq 1$ in Assumption 2. This can be attributed to the assumption that the $A_i$'s commute, in which case the spectral norm of their average governs the bound rather than the maximal norm of the individual $A_i$'s. It is interesting to investigate to what extent our bounds could generalize to the non-commuting case, and if the dependence on $L$ will remain mild. Since our upper bounds make essential use of the AM-GM inequality and of the even more general Maclaurin's inequalities, it seems like we would need strong non-commutative versions of these inequalities to achieve such bounds using our technique. We refer the interested reader to Yun et al. [2021] for further discussion on non-commutative versions of such inequalities.

Another noteworthy observation is regarding the phase transitions in the bounds. In the previously known upper bounds for random reshuffling for quadratics, which scale as $\frac{1}{(nk)^2} + \frac{1}{nk^3}$ (see Table 1), the dominant term switches from $\frac{1}{nk^3}$ to $\frac{1}{(nk)^2}$ once $k \geq n$. In the ill-conditioned setting, our random reshuffling upper bound reveals that such a phase transition only occurs when $k > \frac{\lambda_{\max}}{\lambda} n$, which is a significantly stronger assumption. This suggests that in many real-world applications, the sub-optimality rate would be $\frac{G^2}{\lambda nk} \cdot \min\left\{1\,,\,\frac{\lambda_{\max}^2/\lambda^2}{k^2}\right\}$, unless one can afford a very large iteration budget.

Lastly, we make the following additional remarks on our upper bounds.

**Remark 4** (Bounds with High Probability). *Our proof for the single shuffling upper bound can be adapted to imply a high-probability bound, rather than merely an in-expectation bound, at a cost of a logarithmic dimension dependence (e.g. by taking a union bound over the guarantee in Thm. 5 in the proof across all dimensions). Moreover, in the case where $k \leq \lambda_{\max}/\lambda$, then our random reshuffling upper bound also holds with high probability.[4] This implies a concentration of measure around the expected bounds, which shows that our derived sub-optimality rates are the 'typical' behavior of without-replacement SGD in the worst case for single shuffling, and that the inability to improve upon without replacement when $k \leq \lambda_{\max}/\lambda$ is not just on average but also with high probability for random reshuffling.*

**Remark 5** (The Problem is Always Well-Conditioned in One-Dimension). *Since in the one-dimensional case we have $\lambda = \lambda_{\max}$, our upper bounds reveal that perhaps in contrast to what previous upper bounds that depended on $L$ suggested, the one-dimensional case is always well-conditioned. That is, without-replacement will always beat with-replacement SGD in one-dimension, given that Assumption 2 holds.*

## 5 Experiments

In this section, we empirically verify the phenomenon that our theory predicts by running simulations on the constructions used in Equations (1) and (2).[5] Our lower bounds show that to beat with-replacement on those problem instances, the number of epochs must be in the order of magnitude of the condition number. However, since our analysis is asymptotic in its nature, ignoring constants and logarithmic terms, it is not clear to what extent this phenomenon can be observed in practice. To investigate this, we averaged the performance of 100 SGD instantiations over various values of $k$ ranging from $k = 40$ up to $k = 2,000$, where for each value a suitable step size of $\eta = \frac{\log(nk)}{\lambda nk}$ was chosen. Our problem parameters were chosen to satisfy $n = 500$, $G = 1$, $\lambda = 1$ and $\lambda_{\max} = 200$. Note that this implies that the condition number equals 200 in the constructed problem instances. Each SGD instantiation was initialized from $\mathbf{x}_0 = \left( -\frac{G}{2\lambda}, -\frac{G}{2\lambda_{\max}} \right)$, which satisfies both Assumptions 1 and 2. We used Python 3.6 in our code, which is freely available at https://github.com/ItaySafran/SGD_condition_number. Our code was run on a single machine with an Intel Core i7-8550U 1.80GHz processor.

Our experiment results, which appear in Fig. 1, reveal that to get a noticeable advantage over with-replacement, the number of epochs must be very large and indeed exceed the condition number. Moreover, by that point the remaining error is already extremely tiny (less than $10^{-5}$). Additionally, for values of $k$ up to 300, it is evident that with-replacement sampling occasionally performed almost the same as without-replacement sampling schemes, exhibiting significant overlap of the confidence intervals. This indicates that in certain ill-conditioned applications, it can be quite common for with-replacement to beat without-replacement. Another interesting observation is that random reshuffling does not significantly improve upon single shuffling, unless $k$ is at least $1,000$. This is in line with our theoretical results, which indicate that the advantage of random reshuffling will only be manifest once $k$ is considerably larger than the condition number. It also indicates that the additional effort of reshuffling the functions in every epoch might not pay off on ill-conditioned problems, unless a considerable iteration budget is possible.

We remark that a similar experiment was shown in De Sa [2020], where a construction is presented in which with-replacement outperforms without-replacement SGD. However, a major difference between the above experiment and ours is that in the former, despite under-performing compared to with-replacement, without-replacement achieves an extremely small error, which requires exact arithmetic to monitor, to the point where the difference becomes arguably insignificant from a practical perspective. In contrast, our experiment shows that without-replacement merely does not significantly improve upon with-replacement for moderate values of $k$, in a setting where much more realistic error rates are attained.

---

[4]This is because that in this case, the dominant term in the high probability bound given by $\frac{\lambda_{\max}/\lambda}{k} + \frac{\lambda_{\max}^2/\lambda^2}{k^2}$ is the squared term, which is also the dominant term in the upper bound in Thm. 4.

[5]For the sake of simplicity and to better distinguish the two constructions, we only used the first and third dimensions in the construction in Eq. (2), since the second dimension is identical to the second dimension in Eq. (1).

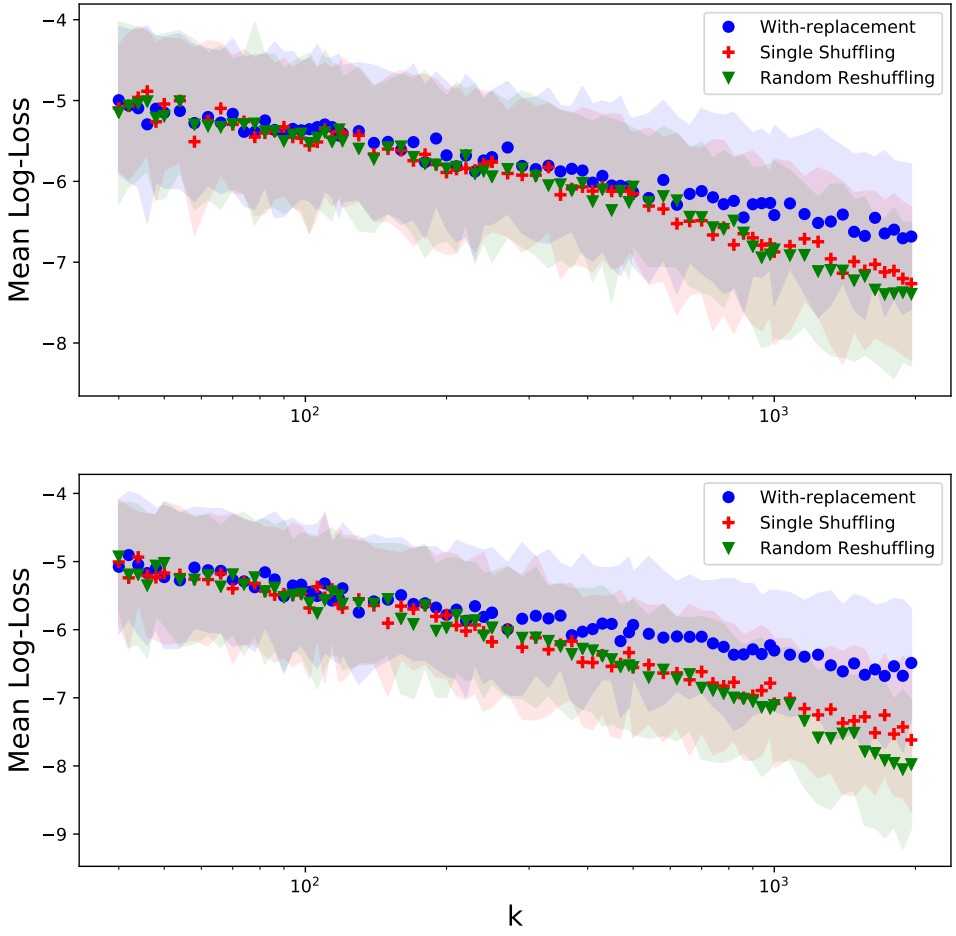

Figure 1: The average value of $\log_{10} F(\mathbf{x}_k)$ attained after running 100 instantiations of SGD on $F(\cdot)$ in Eq. (1) (top) and Eq. (2) (bottom), using with-replacement (blue circle), single shuffling (red plus) and random reshuffling (green triangle) sampling, for varying values of $k$ and where the step size is chosen accordingly. The confidence intervals depict a single standard deviation of the log-loss for each value of $k$. Best viewed in color.

# 6 Discussion

In this paper, we theoretically studied the performance of without-replacement sampling schemes for SGD, compared to the performance of with-replacement SGD. Perhaps surprisingly (in light of previous work on this topic), we showed that without-replacement schemes do not significantly improve over with-replacement sampling, unless the number of epochs $k$ is larger than the condition number, which is unrealistically large in many cases. Although the results are in terms of worst-case bounds, they already hold over a class of rather simple and innocuous quadratic functions. We also showed upper bounds essentially matching our lower bounds, as well as some simple experiments corroborating our findings.

Overall, our results show that the question of how without-replacement schemes compare to with-replacement sampling is intricate, and does not boil down merely to convergence rates in terms of overall number of iterations. Moreover, it re-opens the question of when we can hope without-replacement schemes to have better performance, for moderate values of $k$. Our lower bounds imply that for ill-conditioned problems, this is not possible in the worst-case, already for very simple quadratic constructions. However, an important property of these constructions (as evidenced in Equations (1) and (2)) is that the eigenvalue distribution of the matrices defining the quadratic terms is sharply split between $\lambda_{\max}, \lambda$, and both directions of maximal and minimal curvature are significant. In a sense, this makes it impossible to choose a step size that is optimal for all directions, and $k$ is indeed required to be larger than the condition number $\lambda_{\max}/\lambda$ for without-replacement schemes to beat with-replacement sampling. In contrast, for quadratics whose eigenvalue distribution is smoother (e.g. where the curvature is roughly the same in "most" directions), we expect the "effective" condition number to be smaller, and hence possibly see an improvement already for smaller values of $k$. Formally quantifying this, and finding other favorable cases for without-replacement schemes, is an interesting question for future work. Another open question is to generalize our upper and lower bounds (getting the precise dependence on the condition number) to more general quadratic functions, or the even larger class of strongly-convex and smooth functions.

## Acknowledgments and Disclosure of Funding

This research is supported in part by European Research Council (ERC) Grant 754705. We thank Gilad Yehudai for helpful discussions.

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
