## A  Proofs

### A.1  Proof of Thm. 1

Assume $n$ is even (this is without loss of generality as argued in the beginning of the proof of Thm. 1 in Safran and Shamir [2020]). Recall the function $F$ defined in Eq. (1) by

$$F(\mathbf{x}) \;=\; \frac{1}{n}\sum_{i=1}^{n} f_i(\mathbf{x}) \;=\; \frac{\lambda}{2}x_1^2 + \frac{\lambda_{\max}}{2}x_2^2 \,,$$

where for each $i$,

$$f_i(\mathbf{x}) \;=\; f_i(x_1, x_2) \;:=\; \frac{\lambda}{2}x_1^2 + \frac{\lambda_{\max}}{2}x_2^2 + \begin{cases} \frac{G}{2}x_2 & i \le \frac{n}{2} \\ -\frac{G}{2}x_2 & i > \frac{n}{2} \end{cases} \,.$$

It is readily seen that the above functions satisfy Assumption 1. Assume we initialize at

$$\mathbf{x}_0 = (x_{0,1}, x_{0,2}) = \left( \frac{G}{\lambda} \,, 0 \right) \,,$$

which also satisfies Assumption 1 since $\|\nabla F(\mathbf{x}_0)\| = G$. On these functions, we have that during any single epoch, we perform $n$ iterations of the form

$$x_{new,1} = (1 - \eta\lambda)x_{old,1} \quad,\quad x_{new,2} = (1 - \eta\lambda_{\max})x_{old,2} + \frac{\eta G}{2}\sigma_i \,,$$

where $\sigma_0, \ldots, \sigma_{n-1}$ are a random permutation of $\frac{n}{2}$ 1's and $\frac{n}{2}$ $-1$'s. Repeatedly applying this inequality, we get that after $n$ iterations, the relationship between the first and last iterates in the epoch satisfy

$$x_{t+1,1} \;=\; (1 - \eta\lambda)^n x_{t,1} \quad,\quad x_{t+1,2} \;=\; (1 - \eta\lambda_{\max})^n x_{t,2} + \frac{\eta G}{2}\sum_{i=0}^{n-1}\sigma_i(1 - \eta\lambda_{\max})^{n-i-1} \,.$$

Repeating this across $k$ epochs, we obtain the following relation between the initialization point and what we obtain after $k$ epochs:

$$x_{k,1} = (1-\eta\lambda)^{nk}x_{0,1} \quad,\quad x_{k,2} \;=\; (1-\eta\lambda_{\max})^{nk}x_{0,2} + \frac{\eta G}{2}\cdot\frac{1-(1-\eta\lambda_{\max})^{nk}}{1-(1-\eta\lambda_{\max})^n}\sum_{i=0}^{n-1}\sigma_i(1-\eta\lambda_{\max})^{n-i-1} \,.$$

Noting that $F(\mathbf{x}) = \frac{\lambda}{2}x_1^2 + \frac{\lambda_{\max}}{2}x_2^2$ and $\mathbb{E}[\sigma_i] = 0$, we get that

$$\mathbb{E}[F(\mathbf{x}_k)] \;=\; \frac{\lambda}{2}(1-\eta\lambda)^{2nk}x_{0,1}^2 + \frac{\lambda_{\max}}{2}(1-\eta\lambda_{\max})^{2nk}x_{0,2}^2 + \frac{\eta^2 G^2\lambda_{\max}}{8}\left(\frac{1-(1-\eta\lambda_{\max})^{nk}}{1-(1-\eta\lambda_{\max})^n}\right)^2\beta_{n,\eta,\lambda_{\max}} \,,$$

where

$$\beta_{n,\eta,\lambda_{\max}} \;:=\; \mathbb{E}\left[\left(\sum_{i=0}^{n-1}\sigma_i(1-\lambda_{\max}\eta)^{n-i-1}\right)^2\right] \;=\; \mathbb{E}\left[\left(\sum_{i=0}^{n-1}\sigma_i(1-\lambda_{\max}\eta)^i\right)^2\right] \qquad (3)$$

(using the fact that $\sigma_0, \ldots, \sigma_{n-1}$ are exchangeable random variables). According to Lemma 1 in Safran and Shamir [2020], for some numerical constant $c > 0$,

$$\beta_{n,\eta,\lambda_{\max}} \;\ge\; c\cdot\min\left\{1 + \frac{1}{\lambda_{\max}\eta} \,,\, n^3(\lambda_{\max}\eta)^2\right\} \,. \qquad (4)$$

We now perform a case analysis based on the value of $\eta$:

- If $\eta \le \frac{1}{\lambda nk}$, then

$$\mathbb{E}[F(\mathbf{x}_k)] \;\ge\; \frac{\lambda x_{0,1}^2}{2}(1 - \eta\lambda)^{2nk} \;\ge\; \frac{\lambda x_{0,1}^2}{2}\left(1 - \frac{1}{nk}\right)^{2nk} \;\ge\; \frac{\lambda x_{0,1}^2}{2}\left(\frac{1}{4}\right)^2 \;=\; \frac{\lambda x_{0,1}^2}{32} \,.$$

Substituting $x_{0,1} = G/\lambda$, the above is lower bounded by

$$\frac{G^2}{32\lambda} \,.$$

- If $\eta > \frac{1}{\lambda nk}$ as well as $\eta < \frac{1}{\lambda_{\max}n}$ (assuming this range exists, namely when $k > \lambda_{\max}/\lambda$), then by Bernoulli's inequality we have $(1 - \eta\lambda_{\max})^n \geq 1 - n\eta\lambda_{\max} > 0$, as well as $(1 - \eta\lambda_{\max})^{nk} \leq (1 - \eta\lambda)^{nk} \leq (1 - 1/nk)^{nk} \leq \exp(-1)$, implying that

$$\mathbb{E}[F(\mathbf{x}_k)] \geq \frac{\eta^2 G^2 \lambda_{\max}}{8} \left( \frac{1 - \exp(-1)}{1 - (1 - n\eta\lambda_{\max})} \right)^2 \beta_{n,\eta,\lambda_{\max}} = \frac{\eta^2 G^2 \lambda_{\max}(1 - \exp(-1))^2}{8(n\eta\lambda_{\max})^2} \cdot \beta_{n,\eta,\lambda_{\max}}.$$

Plugging in Eq. (4), and noting that $\eta < \frac{1}{\lambda_{\max}n}$, it is easily verified that $\beta_{n,\eta,\lambda_{\max}} \geq c \cdot \min\{1 + 1/\eta\lambda_{\max}, n^3(\eta\lambda_{\max})^2\} = cn^3\eta^2\lambda_{\max}^2$. This implies that the displayed equation above is at least

$$c' \frac{\eta^2 G^2 \lambda_{\max}}{(n\eta\lambda_{\max})^2} \cdot n^3\eta^2\lambda_{\max}^2 = c'\eta^2 n G^2 \lambda_{\max},$$

for some constant $c'$. Since $\eta > \frac{1}{\lambda nk}$, this is at least

$$c' \frac{n G^2 \lambda_{\max}}{\lambda^2 n^2 k^2} = c' \frac{G^2 \lambda_{\max}}{\lambda^2 n k^2}.$$

- If $\eta > \frac{1}{\lambda nk}$ as well as $\eta \geq \frac{1}{\lambda_{\max}n}$, then noting that $\left( \frac{1 - (1 - \eta\lambda_{\max})^{nk}}{1 - (1 - \eta\lambda_{\max})^n} \right)^2 = \left( \sum_{i=0}^{k-1} ((1 - \eta\lambda_{\max})^n)^i \right)^2 \geq \left( (1 - \eta\lambda_{\max})^0 \right)^2 = 1$ (recall that $n$ is even), we have

$$\mathbb{E}[F(\mathbf{x}_k)] \geq \frac{\eta^2 G^2 \lambda_{\max}}{8} \beta_{n,\eta,\lambda_{\max}}.$$

By the assumption that $\eta \geq \frac{1}{\lambda_{\max}n}$, we have that $n^3(\eta\lambda_{\max})^2 \geq 1/\eta\lambda_{\max}$ as well as $n^3(\eta\lambda_{\max})^2 \geq 1$. Using this and Eq. (4), the above is at least

$$\frac{c\eta^2 G^2 \lambda_{\max}}{8} \min \left\{ 1 + \frac{1}{\eta\lambda_{\max}}, n^3(\eta\lambda_{\max})^2 \right\} \geq \frac{c\eta^2 G^2 \lambda_{\max}}{16} \cdot \left( 1 + \frac{1}{\eta\lambda_{\max}} \right) = \frac{c\eta G^2}{16} (\eta\lambda_{\max} + 1).$$

Since $\eta \geq \frac{1}{\lambda nk}$, this is at least $\frac{cG^2}{16\lambda nk} \left( \frac{\lambda_{\max}}{\lambda nk} + 1 \right)$. Since we assume $\frac{\log(nk)L}{\lambda nk} \leq 1$ which entails $nk \geq \frac{\lambda_{\max}}{\lambda}$, we can further lower bound it (without losing much) by $\frac{cG^2}{16\lambda nk}$.

Combining the cases, we get that regardless of how we choose $\eta$, for some numerical constant $c'' > 0$, it holds that

$$\mathbb{E}[F(\mathbf{x}_k)] \geq c'' \cdot \min \left\{ \frac{G^2}{\lambda nk}, \frac{G^2 \lambda_{\max}}{\lambda^2 n k^2} \right\} = c'' \cdot \frac{G^2}{\lambda nk} \cdot \min \left\{ 1, \frac{\lambda_{\max}/\lambda}{k} \right\}.$$

$\square$

### A.2 Proof of Thm. 2

As in the proof of Thm. 1, we will assume w.l.o.g. that $n$ is even. Recall the function $F$ defined in Eq. (2) by

$$F(\mathbf{x}) = \frac{1}{n} \sum_{i=1}^{n} f_i(\mathbf{x}) = \frac{\lambda}{2} x_1^2 + \frac{\lambda_{\max}}{2} x_2^2 + \frac{\lambda_{\max}}{4} x_3^2,$$

where for each $i$,

$$f_i(\mathbf{x}) = f_i(x_1, x_2, x_3) := \frac{\lambda}{2} x_1^2 + \frac{\lambda_{\max}}{2} x_2^2 + \begin{cases} \frac{G}{2} x_2 + \frac{\lambda_{\max}}{2} x_3^2 + \frac{G}{2} x_3 & i \leq \frac{n}{2} \\ -\frac{G}{2} x_2 - \frac{G}{2} x_3 & i > \frac{n}{2} \end{cases}.$$

Consider the initialization point

$$\mathbf{x}_0 = (x_{0,1}, x_{0,2}, x_{0,3}) = \left( \frac{G}{\lambda}, 0, 0 \right).$$

Note that the above functions satisfy $\|\nabla f_i(\mathbf{x}^*)\| = G/\sqrt{2} \leq G$ for all $i \in [n]$, and that $\|\nabla F(\mathbf{x}_0)\| = G$, therefore Assumption 1 is satisfied. Our proof will analyze the convergence of random reshuffling

SGD under the assumption that $\eta$ belongs to some interval in a partition of the positive real line. For each such interval in the partition, we will take the worst lower bound (i.e. the largest lower bound) along each dimension, where our final lower bound will be the minimum among the bounds derived on each interval.

We begin with deriving an expression for $\mathbf{x}_k$, the iterate after performing $k$ epochs. In a single epoch, after $n$ iterations, the relationship between the first and last iterates in the epoch satisfy

$$
\begin{aligned}
x_{t+1,1} &= (1-\eta\lambda)^n \cdot x_{t,1}\,, \\
x_{t+1,2} &= (1-\eta\lambda_{\max})^n \cdot x_{t,2} + \frac{\eta G}{2}\sum_{i=0}^{n-1}(1-2\sigma_i)(1-\eta\lambda_{\max})^{n-i-1}\,, \\
x_{t+1,3} &= \prod_{i=0}^{n-1}(1-\eta\lambda_{\max}\sigma_i)\cdot x_{t,3} + \frac{\eta G}{2}\sum_{i=0}^{n-1}(1-2\sigma_i)\prod_{j=i+1}^{n-1}(1-\eta\lambda_{\max}\sigma_j)\,, \quad (5)
\end{aligned}
$$

where $\sigma_0,\ldots,\sigma_{n-1}$ are a random permutation of $\frac{n}{2}$ 1's and $\frac{n}{2}$ 0's. Squaring and taking expectation on both sides, using the fact that $\mathbb{E}[1-2\sigma_i]=0$ and that $\mathbf{x}_t$ is independent of the permutation sampled at epoch $t+1$, we have

$$
\begin{aligned}
\mathbb{E}\left[x_{t+1,1}^2\right] &= (1-\eta\lambda)^{2n}x_{t,1}^2\,, \\
\mathbb{E}\left[x_{t+1,2}^2\right] &= (1-\eta\lambda_{\max})^{2n}x_{t,2}^2 + \frac{\eta^2 G^2}{4}\beta_{n,\eta,\lambda_{\max}}\,, \\
\mathbb{E}\left[x_{t+1,3}^2\right] &\geq \mathbb{E}\left[\prod_{i=0}^{n-1}(1-\eta\lambda_{\max}\sigma_i)^2\right]\mathbb{E}\left[x_{t,3}^2\right] + \eta G\mathbb{E}\left[\sum_{i=0}^{n-1}(1-2\sigma_i)\prod_{j=i+1}^{n-1}(1-\eta\lambda_{\max}\sigma_j)\right]\mathbb{E}\left[x_{t,3}\right]\,,
\end{aligned}
$$
$$(6)$$

where $\beta_{n,\eta,\lambda_{\max}}$ is defined in Eq. (3). Unfolding the recursions above for the first two dimensions, we get that after $k$ epochs

$$
\begin{aligned}
\mathbb{E}\left[x_{k,1}^2\right] &= (1-\eta\lambda)^{2nk}x_{0,1}^2\,, \\
\mathbb{E}\left[x_{k,2}^2\right] &= (1-\eta\lambda_{\max})^{2nk}x_{0,2}^2 + \frac{\eta^2 G^2}{4}\cdot\frac{1-(1-\eta\lambda_{\max})^{2nk}}{1-(1-\eta\lambda_{\max})^{2n}}\beta_{n,\eta,\lambda_{\max}}\,.
\end{aligned}
$$

Recalling that $F(\mathbf{x}) = \frac{\lambda}{2}x_1^2 + \frac{\lambda_{\max}}{2}x_2^2 + \frac{\lambda_{\max}}{4}x_3^2 \geq \frac{\lambda}{2}x_1^2 + \frac{\lambda_{\max}}{2}x_2^2$ and combining with the above, we get

$$
\mathbb{E}[F(\mathbf{x}_k)] \geq \frac{\lambda}{2}(1-\eta\lambda)^{2nk}x_{0,1}^2 + \frac{\lambda_{\max}}{2}(1-\eta\lambda_{\max})^{2nk}x_{0,2}^2 + \frac{\lambda_{\max}\eta^2 G^2}{8}\cdot\frac{1-(1-\eta\lambda_{\max})^{2nk}}{1-(1-\eta\lambda_{\max})^{2n}}\beta_{n,\eta,\lambda_{\max}}\,.
$$
$$(7)$$

We now move to a case analysis based on the value of $\eta$:

- If $\eta \leq \frac{1}{\lambda nk}$, we focus on the first dimension. Recall that $x_{0,1} = \frac{G}{\lambda}$ and compute

$$
\begin{aligned}
\mathbb{E}[F(\mathbf{x}_k)] &\geq \frac{\lambda x_{0,1}^2}{2}(1-\eta\lambda)^{2nk} \geq \frac{\lambda x_{0,1}^2}{2}\left(1-\frac{1}{nk}\right)^{2nk} \\
&\geq \frac{\lambda x_{0,1}^2}{2}\left(\frac{1}{4}\right)^2 = \frac{\lambda x_{0,1}^2}{32} = \frac{G^2}{32\lambda} \geq \frac{G^2}{32\lambda nk}\,.
\end{aligned}
$$

- If $\eta > \frac{1}{\lambda nk}$ as well as $\eta < \frac{1}{\lambda_{\max}n}$ (assuming this range exists, namely when $k > \lambda_{\max}/\lambda$), we focus on the second and third dimensions, each resulting in a different dependence on $n,k$. Starting with the second dimension, we have by Bernoulli's inequality that $(1-\eta\lambda_{\max})^{2n} \geq 1-2n\eta\lambda_{\max} > 0$, as well as $(1-\eta\lambda_{\max})^{2nk} \leq (1-\eta\lambda)^{2nk} \leq (1-1/nk)^{2nk} \leq \exp(-2)$, implying that

$$
\mathbb{E}[F(\mathbf{x}_k)] \geq \frac{\eta^2 G^2\lambda_{\max}}{8}\cdot\frac{1-\exp(-2)}{1-(1-2n\eta\lambda_{\max})}\beta_{n,\eta,\lambda_{\max}} = \frac{\eta^2 G^2\lambda_{\max}(1-\exp(-2))}{16n\eta\lambda_{\max}}\cdot\beta_{n,\eta,\lambda_{\max}}\,.
$$

Plugging in Eq. (4), and noting that $\eta < \frac{1}{\lambda_{\max}n}$, it is easily verified that $\beta_{n,\eta,\lambda_{\max}} \geq c \cdot \min\{1 + 1/\eta\lambda_{\max}, n^3(\eta\lambda_{\max})^2\} = cn^3\eta^2\lambda_{\max}^2$. This implies that the displayed equation above is at least

$$c'\frac{\eta^2 G^2\lambda_{\max}}{n\eta\lambda_{\max}} \cdot n^3\eta^2\lambda_{\max}^2 = c'\eta^3 n^2 G^2\lambda_{\max}^2 \,,$$

for some constant $c'$. Since $\eta > \frac{1}{\lambda nk}$, this is at least

$$c'\frac{n^2 G^2\lambda_{\max}^2}{\lambda^3 n^3 k^3} = c'\frac{G^2\lambda_{\max}^2}{\lambda^3 nk^3} \,.$$

Moving to the third dimension, we assume w.l.o.g. that $k \geq n$ (since otherwise the previous bound will be larger than the one to follow). Using Propositions 1 and 2 and Lemma 2 on Eq. (6) (recall that $x_{0,3} = 0$ by our assumption), we have

$$\mathbb{E}\left[x_{t+1,3}^2\right] \geq \left(1 - \frac{\eta\lambda_{\max}n}{2}\right)^2 \mathbb{E}\left[x_{t,3}^2\right] + \frac{\eta^3 G^2\lambda_{\max}n}{128}\left(1 - \left(1 - \frac{\eta\lambda_{\max}n}{2}\right)^t\right)$$

$$\geq (1 - \eta\lambda_{\max}n)\mathbb{E}\left[x_{t,3}^2\right] + \frac{\eta^3 G^2\lambda_{\max}n}{128}\left(1 - \left(1 - \frac{\eta\lambda_{\max}n}{2}\right)^t\right) \,.$$

Unfolding the recursion above yields

$$\mathbb{E}\left[x_{k,3}^2\right] \geq (1 - \eta\lambda_{\max}n)^k x_{0,3}^2 + \frac{\eta^3 G^2\lambda_{\max}n}{128}\sum_{t=0}^{k-1}(1 - \eta\lambda_{\max}n)^{k-t-1}\left(1 - \left(1 - \frac{\eta\lambda_{\max}n}{2}\right)^t\right)$$

$$\geq \frac{\eta^3 G^2\lambda_{\max}n}{128}\sum_{t=\lfloor k/2\rfloor}^{k-1}(1 - \eta\lambda_{\max}n)^{k-t-1}\left(1 - \left(1 - \frac{\eta\lambda_{\max}n}{2}\right)^t\right)$$

$$\geq \frac{\eta^3 G^2\lambda_{\max}n}{128}\sum_{t=\lfloor k/2\rfloor}^{k-1}(1 - \eta\lambda_{\max}n)^{k-t-1}\left(1 - \left(1 - \frac{\eta\lambda_{\max}n}{2}\right)^{\lfloor k/2\rfloor}\right) \,, \tag{8}$$

where we note that the above sum is not empty since we assume $k \geq n \geq 2$. We now have

$$\left(1 - \frac{\eta\lambda_{\max}n}{2}\right)^{\lfloor k/2\rfloor} = \left(1 - \frac{\eta\lambda_{\max}n}{2 \cdot \lfloor k/2\rfloor} \cdot \lfloor k/2\rfloor\right)^{\lfloor k/2\rfloor} \leq \exp\left(-\frac{\eta\lambda_{\max}n\lfloor k/2\rfloor}{2}\right)$$

$$\leq \exp\left(-\frac{\lambda_{\max}\lfloor k/2\rfloor}{2\lambda k}\right) \leq \exp\left(-\frac{\lambda_{\max}}{8\lambda}\right) \leq \exp\left(-\frac{1}{8}\right) \leq 0.9 \,,$$

where the first inequality is due to $(1 - x/y)^y \leq \exp(-x)$ for all $x, y > 0$ and the second inequality is by the assumption $\eta \geq \frac{1}{\lambda nk}$. The above entails

$$1 - \left(1 - \frac{\eta\lambda_{\max}n}{2}\right)^{\lfloor k/2\rfloor} \geq 0.1 \,, \tag{9}$$

which by plugging into Eq. (8) yields

$$\mathbb{E}\left[x_{k,3}^2\right] \geq \frac{\eta^3 G^2\lambda_{\max}n}{1280}\sum_{t=\lfloor k/2\rfloor}^{k-1}(1 - \eta\lambda_{\max}n)^{k-t-1}$$

$$= \frac{\eta^3 G^2\lambda_{\max}n}{1280} \cdot \frac{1 - (1 - \eta\lambda_{\max}n)^{k-\lfloor k/2\rfloor}}{\eta\lambda_{\max}n}$$

$$\geq \frac{\eta^2 G^2}{1280} \cdot \left(1 - \left(1 - \frac{\eta\lambda_{\max}n}{2}\right)^{\lfloor k/2\rfloor}\right) \geq \frac{\eta^2 G^2}{12800} \,,$$

where the last inequality is a second application of Eq. (9). We now conclude with the assumption $\eta \geq \frac{1}{\lambda nk}$ to get

$$\mathbb{E}\left[F(\mathbf{x})\right] \geq \frac{\lambda_{\max}}{4}\mathbb{E}\left[x_{k,3}^2\right] \geq c\frac{G^2\lambda_{\max}}{\lambda^2 n^2 k^2} \,,$$

where $c = \frac{1}{51200}$.

- If $\eta > \frac{1}{\lambda nk}$ as well as $\eta \geq \frac{1}{\lambda_{\max} n}$, we focus on the remainder term of the second dimension in Eq. (7). Noting that $\frac{1-(1-\eta\lambda_{\max})^{2nk}}{1-(1-\eta\lambda_{\max})^{2n}} = \sum_{i=0}^{k-1} \left((1-\eta\lambda_{\max})^{2n}\right)^i \geq (1-\eta\lambda_{\max})^0 = 1$, we have

$$\mathbb{E}[F(\mathbf{x}_k)] \geq \frac{\eta^2 G^2 \lambda_{\max}}{8} \beta_{n,\eta,\lambda_{\max}} .$$

By the assumption that $\eta \geq \frac{1}{\lambda_{\max} n}$, we have that $n^3(\eta\lambda_{\max})^2 \geq 1/\eta\lambda_{\max}$ as well as $n^3(\eta\lambda_{\max})^2 \geq 1$. Using this and Eq. (4), the above is at least

$$\frac{c\eta^2 G^2 \lambda_{\max}}{8} \min\left\{1 + \frac{1}{\eta\lambda_{\max}} , n^3(\eta\lambda_{\max})^2\right\} \geq \frac{c\eta^2 G^2 \lambda_{\max}}{16} \cdot \left(1 + \frac{1}{\eta\lambda_{\max}}\right) = \frac{c\eta G^2}{16}(\eta\lambda_{\max} + 1) .$$

Since $\eta \geq \frac{1}{\lambda nk}$, this is at least $\frac{cG^2}{16\lambda nk}\left(\frac{\lambda_{\max}}{\lambda nk} + 1\right)$. Since we assume $\frac{\log(nk)L}{\lambda nk} \leq 1$ which entails $nk \geq \frac{\lambda_{\max}}{\lambda}$, we can further lower bound it (without losing much) by $\frac{cG^2}{16\lambda nk}$.

$\square$

### A.3 Proof of Thm. 3

Before we prove the theorem, we will first state the following result which handles the one-dimensional case.

**Theorem 5.** *Suppose $F(x) := \frac{\bar{a}}{2}x^2$ and $f_i(x) = \frac{a_i}{2}x^2 - b_i x$, where $\bar{a} = \frac{1}{n}\sum_{i=1}^n a_i$ satisfy Assumption 2, and fix the step size $\eta = \frac{\log(nk)}{\lambda nk}$. Then for any $\delta \in (0,1)$, with probability at least $1-\delta$ over the choice of the permutation $\sigma$, single shuffling SGD satisfies*

$$F(x_k) \leq c \cdot \log^2\left(\frac{8n}{\delta}\right) \cdot \log^2(nk) \cdot \frac{G^2}{\lambda nk} \cdot \min\left\{1 , \frac{\bar{a}/\lambda}{k}\right\} ,$$

*where $c > 0$ is a universal constant. Moreover, this also entails*

$$\mathbb{E}[F(x_k)] \leq \tilde{\mathcal{O}}\left(\frac{G^2}{\lambda nk} \cdot \min\left\{1 , \frac{\bar{a}/\lambda}{k}\right\}\right) ,$$

*where the $\tilde{\mathcal{O}}$ hides a universal constant and factors logarithmic in $n, k, \bar{a}, \lambda$ and their inverses.*

Before we prove the above theorem, we shall first explain why it implies Thm. 3. Since the matrices $A_1, \ldots, A_n$ commute, then they are simultaneously diagonalizable (e.g., Horn and Johnson [2013, Thm. 1.3.21]). Thus, there exists a matrix $P$ such that $P^{-1} A_i P$ is diagonal for all $i \in [n]$. Moreover, since $A_i$ are all symmetric, we may choose such $P$ which is also orthogonal. By Appendix C, we may transform our problem to another quadratic formulation having the same sub-optimality rate and where Assumption 2 is preserved. Following the above reasoning, we may assume w.l.o.g. that $A_i$ is diagonal for all $i \in [n]$.

For some $j \in [d]$ and $t \in [k]$, let $a_j$ and $x_{j,t}$ denote the $j$-th diagonal value of $A$ and $j$-th coordinate of $\mathbf{x}_t$ (the iterate after the $t$-th epoch), respectively. We now explain why we may assume that $\mathbf{b} = \mathbf{0}$. As assumed in Safran and Shamir [2020], mapping $f_i(\mathbf{x}) \mapsto \tilde{f}_i(\mathbf{x} - A^{-1}\mathbf{b})$ for all $i \in [n]$ simply translates our problem so that $\mathbf{x}^* = \mathbf{0}$. By mapping $\mathbf{x}_0$ accordingly, we have that Assumption 2 is preserved, thus we may assume $\mathbf{b} = \mathbf{0}$ w.l.o.g. which entails $\|A_i \mathbf{x}^* - \mathbf{b}_i\| = \|\mathbf{b}_i\| \leq G$ for all $i \in [n]$.

Since we have now reduced our optimization problem to the form $\tilde{f}_i(\mathbf{x}) = \frac{1}{2}\mathbf{x}^\top A_i \mathbf{x} - \mathbf{b}_i^\top \mathbf{x}$ for diagonal $A_i$, we have that the partial derivatives w.r.t. each coordinate are independent of one another, thus we may apply Thm. 5 to each coordinate separately. Letting $F_j(x) = \frac{1}{2}a_j x^2$, we compute

$$\mathbb{E}[F(\mathbf{x}_k)] = \mathbb{E}\left[\frac{1}{2}\mathbf{x}_k^\top A \mathbf{x}_k\right] = \sum_{j=1}^d \mathbb{E}[F_j(x_{j,k})] .$$

Recall that $\mathbf{x}_0 = (x_{1,0}, \ldots, x_{d,0})$. The condition $\|\nabla F(\mathbf{x}_0)\| \leq G$ implies that $\sum_{j=1}^d a_j^2 x_{j,0}^2 \leq G^2$, and since $\lambda \leq a_j \leq \lambda_{\max}$ for all $j \in [d]$, we get $\|\mathbf{x}_0\| \leq \frac{G}{\lambda}$, which in particular implies $x_{j,0} \leq \frac{G}{\lambda}$

for all $j \in [d]$. We now use Thm. 5 applied to each dimension separately and conclude that

$$\mathbb{E}\left[F(\mathbf{x}_k)\right] = \sum_{j=1}^{d} \tilde{\mathcal{O}}\left(\frac{G^2}{\lambda n k} \cdot \min\left\{1, \frac{a_j/\lambda}{k}\right\}\right) \leq \tilde{\mathcal{O}}\left(\frac{G^2}{\lambda n k} \cdot \min\left\{1, \frac{\lambda_{\max}/\lambda}{k}\right\}\right),$$

whereby the $\tilde{\mathcal{O}}$ notation hides a linear term in $d$ which absorbs the sum over the coordinates. $\qquad\square$

*Proof of Thm. 5.* The beginning of the proof is based on deriving a closed-form expression for the iterate at the $k$-th epoch, $x_k$. To this end, we shall use the same derivation as in Safran and Shamir [2020], given here for completeness. First, for a selected permutation $\sigma_i : [n] \to [n]$ we have that the gradient update at iteration $j$ in epoch $i$ is given by

$$x_{new} = \left(1 - \eta a_{\sigma_i(j)}\right) x_{old} + \eta b_{\sigma_i(j)}.$$

Repeatedly applying the above relation, we have that at the end of each epoch the relation between the iterates $x_t$ and $x_{t+1}$ is given by

$$x_{t+1} = \prod_{j=1}^{n}\left(1 - \eta a_{\sigma_{t+1}(j)}\right) x_t + \eta \sum_{j=1}^{n} b_{\sigma_{t+1}(j)} \prod_{i=j+1}^{n}\left(1 - \eta a_{\sigma_{t+1}(i)}\right).$$

Letting

$$S := \prod_{j=1}^{n}\left(1 - \eta a_{\sigma_i(j)}\right) = \prod_{j=1}^{n}\left(1 - \eta a_j\right)$$

and

$$X_{\sigma_t} := \sum_{j=1}^{n} b_{\sigma_t(j)} \prod_{i=j+1}^{n}\left(1 - \eta a_{\sigma_t(i)}\right),$$

this can be rewritten equivalently as

$$x_{t+1} = S x_t + \eta X_{\sigma_{t+1}}. \tag{10}$$

Iteratively applying the above, we have after $k$ epochs that

$$x_k = S^k x_0 + \eta \sum_{i=1}^{k} S^{i-1} X_\sigma = S^k x_0 + \eta \cdot \frac{1 - S^k}{1 - S} X_\sigma. \tag{11}$$

Having derived a closed-form expression for $x_k$, we now turn to make a more careful analysis of the upper bound, improving upon the result of Safran and Shamir [2020]. Note that by our assumptions, $1 \geq 1 - \eta a_j \geq 1 - \eta L \geq 0$ for all $j$, hence $S \in [0, 1]$. As a result, using the fact that $(r + s)^2 \leq 2(r^2 + s^2)$, we have

$$F(x_k) = \frac{\bar{a}}{2} x_k^2 \leq \bar{a}\left(S^{2k} x_0^2 + \eta^2 \left(\frac{1 - S^k}{1 - S}\right)^2 X_\sigma^2\right) \leq \bar{a}\left(S^{2k} \cdot \frac{G^2}{\lambda^2} + \eta^2 \left(\frac{1 - S^k}{1 - S}\right)^2 X_\sigma^2\right)$$

$$\leq S^{2k} \cdot \frac{\bar{a} G^2}{\lambda^2} + \frac{\bar{a}\eta^2}{(1 - S)^2} \cdot X_\sigma^2, \tag{12}$$

whereby $x_0^2 \leq \frac{G^2}{\lambda^2}$ is due to Assumption 2, which entails $|\bar{a} x_0| \leq G$ and thus $|x_0| \leq \frac{G}{\bar{a}} \leq \frac{G}{\lambda}$. We now have

$$S^{2k} = \prod_{j=1}^{n}(1 - \eta a_j)^{2k} \leq (1 - \eta\bar{a})^{2nk} = \left(1 - \frac{\bar{a}\log(nk)}{\lambda n k}\right)^{2nk}$$

$$\leq \exp\left(\frac{-2\bar{a}\log(nk)}{\lambda}\right) = \frac{1}{(nk)^{2\bar{a}/\lambda}} \leq \frac{\lambda}{\bar{a}(nk)^2}, \tag{13}$$

where the first inequality is by the AM-GM inequality applied to $1 - \eta a_1, \ldots, 1 - \eta a_n > 0$, and the last inequality is due to $(nk)^2 \geq 4$ and the fact that $x^y \leq x/y$ for all $x \in [0, 0.25]$ and $y \geq 1$.[6]

---

[6]To see this, we first have that the inequality is trivial when $y = 1$. Assuming $y > 1$, $x/y - x^y$ intersects the $x$ axis iff $x = 0$ or $x = y^{1/(1-y)}$ which is at least $\exp(-1) \geq 0.25$ for $y > 1$, and thus we can verify that $x^y \leq x/y$ for all $x \in [0, \exp(-1)]$ by establishing that $x/y - x^y$ is concave on $(0, \exp(-1))$.

Moreover,

$$S = \prod_{j=1}^{n} (1 - \eta a_j) = \exp\left(\sum_{j=1}^{n} \log(1 - \eta a_j)\right) \leq \exp\left(-\eta \sum_{j=1}^{n} a_j\right) = \exp(-\eta \bar{a} n). \quad (14)$$

Plugging the two displayed equations above into Eq. (12), we get that

$$F(x_k) \leq \frac{G^2}{\lambda(nk)^2} + \frac{\bar{a}\eta^2}{(1 - \exp(-\eta \bar{a} n))^2} \cdot X_\sigma^2. \quad (15)$$

To continue, we will use the following key technical lemma, which we shall use to upper bound $X_\sigma^2$ with high probability (using $\alpha_i := \eta a_i$ and $\beta_i = b_i/G$ for all $i$):

**Lemma 1.** *Let $\alpha_1, \beta_1, \ldots, \alpha_n, \beta_n$ be scalars such that for all $i$, $\alpha_i \in [0,1]$, $|\beta_i| \leq 1$ and $\sum_{i=1}^{n} \beta_i = 0$. Then for any $\delta \in (0,1)$, with probability at least $1 - \delta$, we have*

$$\left(\sum_{j=1}^{n} \beta_{\sigma(j)} \prod_{i=j+1}^{n} (1 - \alpha_{\sigma(i)})\right)^2 \leq c \cdot \log^2\left(\frac{8n}{\delta}\right) \cdot \min\left\{\frac{1}{\bar{\alpha}}, n^3 \bar{\alpha}^2\right\}$$

*where $\bar{\alpha} = \frac{1}{n} \sum_{i=1}^{n} \alpha_i$ and $c > 0$ is a universal constant.*

The proof appears in Subsection B.2. We note that we did not try to optimize the log factor.

**Remark 6.** *This upper bound complements Lemma 1 from Safran and Shamir [2020], which analyzed the same key quantity in the special case where $\beta_i \in \{-1,+1\}$ and $\alpha_i = \bar{\alpha}$ are the same for all $i$, and showed (when $\bar{\alpha} \in [0,1]$) a __lower bound__ of $c' \cdot \min\left\{\frac{1}{\bar{\alpha}}, n^3 \bar{\alpha}^2\right\}$ for some universal constant $c' > 0$.[7] This implies that our upper bound is tight up to constants and logarithmic factors.*

We now consider two cases, depending on the value of $\eta \bar{a} n$:

- **Case 1:** $\eta \bar{a} n \leq \frac{1}{2}$. By Lemma 1, with probability at least $1 - \delta$,

$$X_\sigma^2 \leq c \cdot \log^2(8n/\delta) \cdot G^2 n^3 (\eta \bar{a})^2.$$

In addition,

$$\exp(-\eta \bar{a} n) \leq 1 - \frac{1}{2} \eta \bar{a} n,$$

due to the assumption $\eta \bar{a} n \leq \frac{1}{2}$ and the fact that $\exp(-z) \leq 1 - \frac{1}{2}z$ for all $z \in [0, 1/2]$. Plugging the two displayed equations above back into Eq. (15), we get that with probability at least $1 - \delta$,

$$
\begin{aligned}
F(x_k) &\leq \frac{G^2}{\lambda nk^2} + \frac{\bar{a}\eta^2}{(\eta \bar{a} n/2)^2} \cdot c \cdot \log^2(8n/\delta) \cdot G^2 n^3 (\eta \bar{a})^2 \\
&= \frac{G^2}{\lambda(nk)^2} + 4c \cdot \log^2(8n/\delta) \cdot \bar{a} n G^2 \eta^2 \\
&= \frac{G^2}{\lambda(nk)^2} + 4c \cdot \log^2(8n/\delta) \cdot \bar{a} n G^2 \cdot \left(\frac{\log(nk)}{\lambda nk}\right)^2 \\
&= \frac{G^2}{\lambda(nk)^2} + 4c \cdot \log^2(8n/\delta) \cdot \log^2(nk) \cdot \frac{\bar{a} G^2}{\lambda^2 nk^2} \\
&\leq \left(1 + 4c \cdot \log^2(8n/\delta) \cdot \log^2(nk)\right) \cdot \frac{\bar{a} G^2}{\lambda^2 nk^2},
\end{aligned}
$$

where in the last step we used the fact that $\frac{1}{n} \leq 1 \leq \frac{\bar{a}}{\lambda}$. Likewise, bounding $X_\sigma^2$ in expectation using Proposition 3 yields a bound of

$$\mathbb{E}[F(x_k)] \leq \tilde{\mathcal{O}}\left(\frac{\bar{a} G^2}{\lambda^2 nk^2}\right).$$

---

[7]In Safran and Shamir [2020], the exact lower bound is $c' \min\left\{1 + \frac{1}{\bar{\alpha}}, n^3 \bar{\alpha}^2\right\}$, which is equivalent to $c'' \min\left\{\frac{1}{\bar{\alpha}}, n^3 \bar{\alpha}^2\right\}$ for some constant $c'' > 0$

- **Case 2:** $\eta\bar{a}n > \frac{1}{2}$. By Lemma 1, with probability at least $1 - \delta$,

$$X_\sigma^2 \ \leq \ \frac{c\log^2(8n/\delta)}{\eta\bar{a}} \ .$$

In addition, $\eta\bar{a}n = \frac{\log(nk)\bar{a}}{\lambda k} > \frac{1}{2}$. Plugging these back into Eq. (15), we get that

$$
\begin{aligned}
F(x_k) \ &\leq \ \frac{G^2}{\lambda(nk)^2} + \frac{\bar{a}\eta^2}{(1-\exp(-1/2))^2} \cdot \frac{cG^2\log^2(8n/\delta)}{\eta\bar{a}} \\
&\leq \ \frac{G^2}{\lambda(nk)^2} + \frac{cG^2\log^2(8n/\delta)}{(1-\exp(-1/2))^2} \cdot \eta \\
&= \ \frac{G^2}{\lambda(nk)^2} + \frac{c\log^2(8n/\delta)\cdot\log(nk)}{(1-\exp(-1/2))^2} \cdot \frac{G^2}{\lambda nk} \\
&\leq \ \left(1 + \frac{c\log^2(8n/\delta)\cdot\log(nk)}{(1-\exp(-1/2))^2}\right) \cdot \frac{G^2}{\lambda nk} \ .
\end{aligned}
$$

Likewise, bounding $X_\sigma^2$ in expectation using Proposition 3 yields a bound of

$$\mathbb{E}\left[F(x_k)\right] \ \leq \ \tilde{\mathcal{O}}\left(\frac{G^2}{\lambda nk}\right) \ .$$

To combine the two cases, we note that the condition $\eta\bar{a}n \leq \frac{1}{2}$ is equivalent to $\frac{\bar{a}\log(nk)}{\lambda k} \leq \frac{1}{2}$. In that case, we have $\frac{\log(nk)\bar{a}G^2}{\lambda^2 nk^2} \ \leq \ \frac{1}{2} \cdot \frac{G^2}{\lambda nk}$, and thus

$$\frac{\log(nk)\bar{a}G^2}{\lambda^2 nk^2} \ \leq \ \min\left\{\frac{\log(nk)\bar{a}G^2}{\lambda^2 nk^2} \ , \ \frac{G^2}{2\lambda nk}\right\} \ = \ \frac{G^2}{\lambda nk} \cdot \min\left\{\frac{\log(nk)\bar{a}}{\lambda k} \ , \ \frac{1}{2}\right\} \ .$$

In the opposite case where $\frac{\bar{a}\log(nk)}{\lambda k} > \frac{1}{2}$, it follows that $\frac{G^2}{\lambda nk} \ < \ \frac{2\log(nk)\bar{a}G^2}{\lambda^2 nk^2}$, and therefore

$$\frac{G^2}{\lambda nk} \ \leq \ \min\left\{\frac{2\log(nk)\bar{a}G^2}{\lambda^2 nk^2} \ , \ \frac{G^2}{\lambda nk}\right\} \ = \ \frac{G^2}{\lambda nk} \cdot \min\left\{\frac{2\log(nk)\bar{a}}{\lambda k} \ , \ 1\right\} \ .$$

Plugging these two inequalities into the bounds obtained in the two cases above and simplifying a bit, the result follows. $\qquad\square$

### A.4 Proof of Thm. 4

Similarly to the proof of Thm. 3, we first assume w.l.o.g. that $A_i$ is diagonal for all $i \in [n]$ and that $\mathbf{b} = \mathbf{0}$, implying a per-coordinate gradient bound of $G$ (see the argument following Thm. 5 for justification). Under the same reasoning, the proof then follows from the following theorem. $\qquad\square$

**Theorem 6.** *Suppose* $F(x) := \frac{\bar{a}}{2}x^2$ *and* $f_i(x) = \frac{a_i}{2}x^2 - b_i x$, *where* $\bar{a} = \frac{1}{n}\sum_{i=1}^{n} a_i$ *satisfy Assumption 2, and fix the step size* $\eta = \frac{\log(nk)}{\lambda nk}$. *Then random reshuffling SGD satisfies*

$$\mathbb{E}\left[F(x_k)\right] \ \leq \ \tilde{\mathcal{O}}\left(\frac{G^2}{\lambda nk} \cdot \min\left\{1 \ , \ \frac{\bar{a}/\lambda}{nk} + \frac{\bar{a}^2/\lambda^2}{k^2}\right\}\right) \ .$$

*where the* $\tilde{\mathcal{O}}$ *hides a universal constant and factors logarithmic in* $n, k, \bar{a}, \lambda$ *and their inverses.*

*Proof.* Our analysis picks off from Safran and Shamir [2020, Eq. (22)]. However, for the sake of completeness we shall include the derivation of Eq. (22) as was done in the above reference.

Continuing from Eq. (10), we square and take expectation on both sides to obtain

$$\mathbb{E}\left[x_{t+1}^2\right] \ = \ \mathbb{E}\left[\left(Sx_t + \eta X_{\sigma_{t+1}}\right)^2\right] = S^2\mathbb{E}[x_t^2] + 2\eta S\mathbb{E}\left[x_t X_{\sigma_{t+1}}\right] + \eta^2\mathbb{E}\left[X_{\sigma_{t+1}}^2\right] \ .$$

Since in random reshuffling the random component at iteration $t+1$, $X_{\sigma_{t+1}}$, is independent of the iterate at iteration $t$, $x_t$, and by plugging $t = k$ into Eq. (11), the above equals

$$
\begin{aligned}
\mathbb{E}\left[x_{t+1}^2\right] &= S^2\mathbb{E}[x_t^2] + 2\eta S\mathbb{E}\left[x_t\right]\mathbb{E}\left[X_{\sigma_{t+1}}\right] + \eta^2\mathbb{E}\left[X_{\sigma_{t+1}}^2\right] \\
&= S^2\mathbb{E}[x_t^2] + 2\eta S\mathbb{E}\left[S^t x_0 + \eta\sum_{i=1}^{t} S^{t-i}X_{\sigma_i}\right]\mathbb{E}\left[X_{\sigma_{t+1}}\right] + \eta^2\mathbb{E}\left[X_{\sigma_{t+1}}^2\right] \\
&= S^2\mathbb{E}[x_t^2] + 2\eta S^{t+1}x_0\mathbb{E}\left[X_{\sigma_{t+1}}\right] + 2\eta^2\sum_{i=1}^{t} S^{t-i+1}\mathbb{E}\left[X_{\sigma_i}\right]\mathbb{E}\left[X_{\sigma_{t+1}}\right] + \eta^2\mathbb{E}\left[X_{\sigma_{t+1}}^2\right] \\
&= S^2\mathbb{E}[x_t^2] + 2\eta S^{t+1}x_0\mathbb{E}\left[X_{\sigma_1}\right] + 2\eta^2\sum_{i=1}^{t} S^{t-i+1}\mathbb{E}\left[X_{\sigma_1}\right]^2 + \eta^2\mathbb{E}\left[X_{\sigma_1}^2\right] ,
\end{aligned}
$$

where the last equality is due to $X_{\sigma_i}$ being i.i.d. for all $i$. Recursively applying the above relation and taking absolute value, we obtain

$$
\mathbb{E}\left[x_k^2\right] = S^{2k}x_0^2 + 2\eta x_0\mathbb{E}\left[X_{\sigma_1}\right]\sum_{j=0}^{k-1}S^{k+j} + 2\eta^2\mathbb{E}\left[X_{\sigma_1}\right]^2\sum_{j=0}^{k-1}S^{2j}\sum_{i=1}^{k-j-1}S^i + \eta^2\mathbb{E}\left[X_{\sigma_1}^2\right]\sum_{j=0}^{k-1}S^{2j}.
$$

Having derived the bound appearing in Safran and Shamir [2020, Eq. 22], we now turn to improve their result by refining the upper bound as follows. We have that the above implies

$$
\begin{aligned}
\mathbb{E}\left[x_k^2\right] &= S^{2k}x_0^2 + 2\eta x_0\mathbb{E}\left[X_{\sigma_1}\right]S^k\frac{1-S^k}{1-S} + 2\eta^2\mathbb{E}\left[X_{\sigma_1}\right]^2 S\sum_{j=0}^{k-1}S^{2j}\cdot\frac{1-S^{k-j-1}}{1-S} + \eta^2\mathbb{E}\left[X_{\sigma_1}^2\right]\frac{1-S^{2k}}{1-S} \\
&\leq S^{2k}x_0^2 + 2\eta x_0\mathbb{E}\left[X_{\sigma_1}\right]S^k\frac{1}{1-S} + 2\eta^2\mathbb{E}\left[X_{\sigma_1}\right]^2\sum_{j=0}^{k-1}S^{2j}\cdot\frac{1}{1-S} + \eta^2\mathbb{E}\left[X_{\sigma_1}^2\right]\frac{1}{1-S} \\
&\leq S^{2k}x_0^2 + 2\eta x_0\mathbb{E}\left[X_{\sigma_1}\right]S^k\frac{1}{1-S} + 2\eta^2\mathbb{E}\left[X_{\sigma_1}\right]^2\frac{1-S^{2k}}{(1-S)^2} + \eta^2\mathbb{E}\left[X_{\sigma_1}^2\right]\frac{1}{1-S} \\
&\leq S^{2k}x_0^2 + 2\eta x_0\mathbb{E}\left[X_{\sigma_1}\right]S^k\frac{1}{1-S} + 2\eta^2\mathbb{E}\left[X_{\sigma_1}\right]^2\frac{1}{(1-S)^2} + \eta^2\mathbb{E}\left[X_{\sigma_1}^2\right]\frac{1}{1-S} .
\end{aligned}
$$

Using the fact that $(r+s)^2 \leq 2(r^2+s^2)$ for $r = S^k x_0$ and $s = \eta\mathbb{E}\left[X_{\sigma_1}^2\right]/(1-S)$, we have

$$
\mathbb{E}\left[x_k^2\right] \leq 2S^{2k}x_0^2 + 3\eta^2\mathbb{E}\left[X_{\sigma_1}\right]^2\frac{1}{(1-S)^2} + \eta^2\mathbb{E}\left[X_{\sigma_1}^2\right]\frac{1}{1-S} .
$$

Next, we use the assumption $x_0^2 \leq \frac{G^2}{\lambda^2}$ (which follows from Assumption 2, since it entails $|\bar{a}x_0| \leq G$ and thus $|x_0| \leq \frac{G}{\bar{a}} \leq \frac{G}{\lambda}$), along with Equations (13) and (14) to upper bound the above by

$$
\frac{2G^2}{\bar{a}\lambda n^2 k^2} + \frac{3\eta^2\mathbb{E}\left[X_{\sigma_1}\right]^2}{(1-\exp(-\eta\bar{a}n))^2} + \frac{\eta^2\mathbb{E}\left[X_{\sigma_1}^2\right]}{1-\exp(-\eta\bar{a}n)} . \tag{16}
$$

We now consider two cases, depending on the value of $\eta\bar{a}n$, using Proposition 3 by letting $\alpha_j = \eta a_j$ and $\beta_j = b_j/G$:

- **Case 1:** $\eta\bar{a}n \leq \frac{1}{2}$. We have that Eq. (16) is upper bounded by

$$
\begin{aligned}
&\frac{2G^2}{\bar{a}\lambda n^2 k^2} + \frac{12G^2\eta^4 n^2\bar{a}^2}{(1-\exp(-\eta\bar{a}n))^2} + c_2\log^2\left(\frac{8}{\eta n\bar{a}}\right)\cdot\frac{G^2\eta^4 n^3\bar{a}^2}{1-\exp(-\eta\bar{a}n)} \\
&\leq \frac{2G^2}{\bar{a}\lambda n^2 k^2} + 48G^2\eta^2 + 2c_2\log^2\left(\frac{8}{\eta n\bar{a}}\right)\cdot G^2\eta^3 n^2\bar{a} \\
&\leq \tilde{\mathcal{O}}\left(\frac{G^2}{\bar{a}\lambda n^2 k^2} + \frac{G^2}{\lambda^2 n^2 k^2} + \frac{G^2\bar{a}}{\lambda^3 n k^3}\right) \leq \tilde{\mathcal{O}}\left(\frac{G^2}{\lambda^2 n^2 k^2} + \frac{G^2\bar{a}}{\lambda^3 n k^3}\right) ,
\end{aligned}
$$

where we used the inequality $\exp(-x) \le 1 - x/2$ which holds for all $x \in [0, 1/2]$, and the fact that $\frac{\bar{a}}{\lambda} \ge 1$. Using the definition of $F$ we get

$$\mathbb{E}\left[F(x_k)\right] \;=\; \frac{\bar{a}}{2}\mathbb{E}\left[x_k^2\right] \;\le\; \tilde{\mathcal{O}}\left(\frac{G^2\bar{a}}{\lambda^2 n^2 k^2} + \frac{G^2\bar{a}^2}{\lambda^3 n k^3}\right).$$

- **Case 2:** $\eta\bar{a}n > \frac{1}{2}$. In this case we have that Eq. (16) is upper bounded by

$$\frac{2G^2}{\bar{a}\lambda n^2 k^2} + c_1^2 \log^2\left(\sqrt{2\bar{\alpha}}\cdot 8n^2\right)\frac{3G^2\eta^2}{\eta\bar{a}(1-\exp(-1/2))^2} + c_3^2 \log^2\left(8n^2\bar{\alpha}^2\right)\cdot \frac{G^2\eta^2}{\eta\bar{a}(1-\exp(-1/2))}$$

$$\le\; \tilde{\mathcal{O}}\left(\frac{G^2}{\bar{a}\lambda n^2 k^2} + \frac{G^2\eta}{\bar{a}}\right) \;\le\; \tilde{\mathcal{O}}\left(\frac{G^2}{\bar{a}\lambda n^2 k^2} + \frac{G^2}{\bar{a}\lambda nk}\right) \;\le\; \tilde{\mathcal{O}}\left(\frac{G^2}{\bar{a}\lambda nk}\right).$$

Using the definition of $F$ we get

$$\mathbb{E}\left[F(x_k)\right] \;=\; \frac{\bar{a}}{2}\mathbb{E}\left[x_k^2\right] \;\le\; \tilde{\mathcal{O}}\left(\frac{G^2}{\lambda nk}\right).$$

To combine the two cases, we note that the condition $\eta\bar{a}n \le \frac{1}{2}$ is equivalent to $\frac{\bar{a}\log(nk)}{\lambda k} \le \frac{1}{2}$. In that case we have

$$\frac{\log(nk)\bar{a}G^2}{\lambda^2 n^2 k^2} + \frac{\log^2(nk)\bar{a}^2 G^2}{\lambda^3 n k^3} \;\le\; \frac{G^2}{2\lambda nk} + \frac{G^2}{4\lambda nk} \;\le\; \frac{G^2}{\lambda nk},$$

implying

$$\frac{\log(nk)\bar{a}G^2}{\lambda^2 n^2 k^2} + \frac{\log^2(nk)\bar{a}^2 G^2}{\lambda^3 n k^3} \;\le\; \min\left\{\frac{\log(nk)\bar{a}G^2}{\lambda^2 n^2 k^2} + \frac{\log^2(nk)\bar{a}^2 G^2}{\lambda^3 n k^3}\,,\, \frac{G^2}{\lambda nk}\right\}$$

$$=\; \frac{G^2}{\lambda nk}\cdot\min\left\{1\,,\, \frac{\log(nk)\bar{a}}{\lambda nk} + \frac{\log^2(nk)\bar{a}^2}{\lambda^2 k^2}\right\}.$$

In the opposite case where $\frac{\bar{a}\log(nk)}{\lambda k} > \frac{1}{2}$, it follows that $1 < \frac{2\bar{a}\log(nk)}{\lambda k}$, and therefore

$$\frac{G^2}{\lambda nk} \;\le\; \frac{4\bar{a}^2 \log^2(nk)G^2}{\lambda^3 n k^3} \;\le\; \frac{4\bar{a}\log(nk)G^2}{\lambda^2 n^2 k^2} + \frac{4\bar{a}^2 \log^2(nk)G^2}{\lambda^3 n k^3}$$

$$=\; 4\frac{G^2}{\lambda nk}\cdot\min\left\{1\,,\, \frac{\log(nk)\bar{a}}{\lambda nk} + \frac{\log^2(nk)\bar{a}^2}{\lambda^2 k^2}\right\}.$$

Plugging these two inequalities into the bounds obtained in the two cases above and absorbing logarithmic terms into the big $\tilde{\mathcal{O}}$ notation, the result follows. $\qquad\square$

## B  Technical Lemmas

### B.1  Proofs of Propositions

**Proposition 1.** *Suppose $\sigma_0, \ldots, \sigma_{n-1}$ is a random permutation of $\frac{n}{2}$ 0's and $\frac{n}{2}$ 1's and $\eta \le \frac{1}{\lambda_{\max}n}$. Then*

$$\mathbb{E}\left[\left(\prod_{i=0}^{n-1}(1 - \eta\lambda_{\max}\sigma_i)\right)\left(\sum_{i=0}^{n-1}(1-2\sigma_i)\prod_{j=i+1}^{n-1}(1-\eta\lambda_{\max}\sigma_j)\right)\right] \;\le\; -\frac{1}{16}\eta\lambda_{\max}n.$$

*Proof.* Starting with the first multiplicand, we have from Lemma 2 that it is lower bounded by $0.5$ deterministically, as it does not depend on the permutation sampled, thus we can take it outside the expectation. At this point, the statement in the proposition reduces to showing that

$$\mathbb{E}\left[\sum_{i=0}^{n-1}(1-2\sigma_i)\prod_{j=i+1}^{n-1}(1-\eta\lambda_{\max}\sigma_j)\right] \;\le\; -\frac{1}{8}\eta\lambda_{\max}n,$$

which follows immediately from Lemma 4. $\qquad\square$

**Proposition 2.** *Suppose $\sigma_0, \ldots, \sigma_{n-1}$ is a random permutation of $\frac{n}{2}$ 0's and $\frac{n}{2}$ 1's and $\eta \leq \frac{1}{\lambda_{\max} n}$. Let*

$$x_{t+1} = \prod_{i=0}^{n-1}(1 - \eta\lambda_{\max}\sigma_i) \cdot x_t + \frac{\eta G}{2}\sum_{i=0}^{n-1}(1 - 2\sigma_i)\prod_{j=i+1}^{n-1}(1 - \eta\lambda_{\max}\sigma_j),$$

*where $x_0 = 0$. Then*

$$\mathbb{E}\left[x_k\right] \leq -\frac{\eta G}{8}\left(1 - \left(1 - \frac{\eta\lambda_{\max}n}{2}\right)^k\right).$$

*Proof.* Taking expectation on both sides, using the fact that the iterate at the $t$-th epoch, $x_t$ is independent of the permutation sampled at epoch $t + 1$, we have

$$\mathbb{E}\left[x_{t+1}\right] = \mathbb{E}\left[x_t\right]\mathbb{E}\left[\prod_{i=0}^{n-1}(1 - \eta\lambda_{\max}\sigma_i)\right] + \frac{\eta G}{2}\mathbb{E}\left[\sum_{i=0}^{n-1}(1 - 2\sigma_i)\prod_{j=i+1}^{n-1}(1 - \eta\lambda_{\max}\sigma_j)\right].$$

Recall that $x_0 = 0$. Using Lemmas 2 and 4, we have by a simple inductive argument that $\mathbb{E}\left[x_t\right] \leq 0$ for all $t$, and therefore

$$\mathbb{E}\left[x_{t+1}\right] \leq \left(1 - \frac{\eta\lambda_{\max}n}{2}\right)\mathbb{E}\left[x_t\right] - \frac{\eta^2\lambda_{\max}nG}{16}.$$

Unfolding the above recursion, we have

$$\mathbb{E}\left[x_k\right] \leq \left(1 - \frac{\eta\lambda_{\max}n}{2}\right)^k x_0 - \frac{\eta^2\lambda_{\max}nG}{16}\sum_{i=0}^{k-1}\left(1 - \frac{\eta\lambda_{\max}n}{2}\right)^i$$

$$= -\frac{\eta^2\lambda_{\max}nG}{16}\sum_{i=0}^{k-1}\left(1 - \frac{\eta\lambda_{\max}n}{2}\right)^i = -\frac{\eta^2\lambda_{\max}nG}{16} \cdot \frac{1 - \left(1 - \frac{\eta\lambda_{\max}n}{2}\right)^k}{0.5\eta\lambda_{\max}n}$$

$$= -\frac{\eta G}{8}\left(1 - \left(1 - \frac{\eta\lambda_{\max}n}{2}\right)^k\right).$$

$\square$

**Proposition 3.** *Let $\alpha_1, \beta_1, \ldots, \alpha_n, \beta_n$ be scalars such that for all $i$, $\alpha_i \in [0,1]$, $|\beta_i| \leq 1$ and $\sum_{i=1}^n \beta_i = 0$. Then*

- 
$$\mathbb{E}\left[\left|\sum_{j=1}^n \beta_{\sigma(j)}\prod_{i=j+1}^n(1 - \alpha_{\sigma(i)})\right|\right] \leq \begin{cases} 2n\bar{\alpha} & n\bar{\alpha} \leq \frac{1}{2} \\ c_1\frac{\log\left(\sqrt{2\bar{\alpha}}\cdot 8n^2\right)}{\sqrt{\bar{\alpha}}} & n\bar{\alpha} > \frac{1}{2} \end{cases},$$

- 
$$\mathbb{E}\left[\left(\sum_{j=1}^n \beta_{\sigma(j)}\prod_{i=j+1}^n(1 - \alpha_{\sigma(i)})\right)^2\right] \leq \begin{cases} c_2 \cdot \log^2\left(\frac{8}{n\bar{\alpha}}\right)\cdot n^3\bar{\alpha}^2 & n\bar{\alpha} \leq \frac{1}{2} \\ c_3 \cdot \frac{\log^2\left(8n^2\bar{\alpha}^2\right)}{\bar{\alpha}} & n\bar{\alpha} > \frac{1}{2} \end{cases},$$

*where $\bar{\alpha} = \frac{1}{n}\sum_{i=1}^n \alpha_i$ and $c_1, c_2, c_3 > 0$ are universal constants.*

*Proof.* The first part of the proof focuses on bounding the first term in absolute value for the case $\frac{1}{\bar{\alpha}} \leq n^3\bar{\alpha}^2$. It is a minor refinement of Lemma 8 in Safran and Shamir [2020]. Define

$$Y_j := \beta_{\sigma(j)}\prod_{i=j+1}^n(1 - \alpha_{\sigma(i)}). \tag{17}$$

Assuming $n\bar{\alpha} \leq \frac{1}{2}$, we have from Safran and Shamir [2020, Equations (31),(32)] that

$$\mathbb{E}[Y_j] = \sum_{m=1}^{n-j} (-1)^m \mathbb{E}\left[\beta_{\sigma(j)} \sum_{j+1 \leq i_1,\ldots,i_m \leq n \text{ distinct}} \prod_{l=1}^{m} \alpha_{\sigma(i_l)}\right], \tag{18}$$

and

$$\mathbb{E}\left[\beta_{\sigma(j)} \sum_{j+1 \leq i_1,\ldots,i_m \leq n \text{ distinct}} \prod_{l=1}^{m} \alpha_{\sigma(i_l)}\right]$$

$$= -\frac{(n-m)!}{n!} \sum_{t_1 \in [n]} \sum_{t_2 \in [n]\setminus\{t_1\}} \cdots \sum_{t_m \in [n]\setminus\{t_1,\ldots,t_{m-1}\}} \alpha_{t_1}\alpha_{t_2}\ldots\alpha_{t_m} \frac{1}{n-m} \sum_{t_{m+1} \in \{t_1,\ldots,t_m\}} \beta_{t_{m+1}}$$

$$= -\frac{1}{n-m}\binom{n}{m}^{-1} \sum_{1 \leq t_1 < \ldots < t_m \leq n} \alpha_{t_1}\alpha_{t_2}\ldots\alpha_{t_m} \sum_{t_{m+1} \in \{t_1,\ldots,t_m\}} \beta_{t_{m+1}} .$$

Using Maclaurin's inequality and the assumption $\beta_j \leq 1$ for all $j$, the above is upper bounded in absolute value by

$$\frac{m}{n-m}\bar{\alpha}^m .$$

Plugging this in Eq. (17) yields

$$|\mathbb{E}[Y_j]| \leq \sum_{m=1}^{n-j} \left|(-1)^m \frac{m}{n-m}\bar{\alpha}^m\right| \leq \sum_{m=1}^{n-1} \frac{m}{n-m}\bar{\alpha}^m$$

$$\leq \sum_{m=1}^{n-1} n^{m-1}\bar{\alpha}^m = \bar{\alpha}\frac{1-(\bar{\alpha}n)^{n-1}}{1-\bar{\alpha}n} \leq 2\bar{\alpha} ,$$

where the third inequality is due to $n \geq 2$ and the last inequality is by the assumption $n\bar{\alpha} \leq \frac{1}{2}$. Lastly, we plug the above in Eq. (18) to obtain

$$\mathbb{E}\left[\left|\sum_{j=1}^{n} \beta_{\sigma(j)} \prod_{i=j+1}^{n} (1-\alpha_{\sigma(i)})\right|\right] \leq 2n\bar{\alpha} . \tag{19}$$

Assuming $n\bar{\alpha} > \frac{1}{2}$, we have from Lemma 1 with probability at least $1 - \frac{1}{n\sqrt{2\bar{\alpha}}} > 0$ that

$$\left(\sum_{j=1}^{n} \beta_{\sigma(j)} \prod_{i=j+1}^{n} (1-\alpha_{\sigma(i)})\right)^2 \leq c \cdot \log^2\left(\sqrt{2\bar{\alpha}} \cdot 8n^2\right) \cdot \min\left\{\frac{1}{\bar{\alpha}} , n^3\bar{\alpha}^2\right\}$$

$$\leq c \cdot \log^2\left(\sqrt{2\bar{\alpha}} \cdot 8n^2\right) \cdot \frac{1}{\bar{\alpha}} . \tag{20}$$

Compute using the law of total expectation and the square root of the above equation, using the fact that the square root of the above quantity is deterministically upper bounded by $n$ due to the assumptions $\beta_j \leq 1$ and $\alpha_j \leq 1$

$$\mathbb{E}\left[\left|\sum_{j=1}^{n} \beta_{\sigma(j)} \prod_{i=j+1}^{n} (1-\alpha_{\sigma(i)})\right|\right] \leq n \cdot \frac{1}{n\sqrt{2\bar{\alpha}}} + \sqrt{c} \cdot \log\left(\sqrt{2\bar{\alpha}} \cdot 8n^2\right) \cdot \frac{1}{\sqrt{\bar{\alpha}}}\left(1 - \frac{1}{n\sqrt{2\bar{\alpha}}}\right)$$

$$\leq \cdot \frac{1}{\sqrt{2\bar{\alpha}}} + \log\left(\sqrt{2\bar{\alpha}} \cdot 8n^2\right) \cdot \frac{\sqrt{c}}{\sqrt{\bar{\alpha}}} \leq \frac{c_1 \log\left(\sqrt{2\bar{\alpha}} \cdot 8n^2\right)}{\sqrt{\bar{\alpha}}} ,$$

for some constant $c_1 > 0$. Combining the above with Eq. (19) completes the first part of the proposition. Moving to the second assuming $n\bar{\alpha} \leq \frac{1}{2}$, we have again from Lemma 1 with probability at least $1 - n\bar{\alpha}^2 > 0$ that

$$\left(\sum_{j=1}^{n} \beta_{\sigma(j)} \prod_{i=j+1}^{n} (1-\alpha_{\sigma(i)})\right)^2 \leq c \cdot \log^2\left(\frac{8}{n\bar{\alpha}}\right) \cdot \min\left\{\frac{1}{\bar{\alpha}} , n^3\bar{\alpha}^2\right\} .$$

From the law of total expectation, the above, and the fact that the quantity above is deterministically upper bounded by $n^2$ due to the assumptions $\beta_j \leq 1$ and $\alpha_j \leq 1$, we have

$$\mathbb{E}\left[\left(\sum_{j=1}^{n}\beta_{\sigma(j)}\prod_{i=j+1}^{n}(1-\alpha_{\sigma(i)})\right)^2\right] \leq n^2 \cdot n\bar{\alpha}^2 + c \cdot \log^2\left(\frac{8}{n\bar{\alpha}}\right) \cdot \min\left\{\frac{1}{\bar{\alpha}}, n^3\bar{\alpha}^2\right\} \cdot \left(1 - n\bar{\alpha}^2\right)$$

$$\leq n^3\bar{\alpha}^2 + c \cdot \log^2\left(\frac{8}{n\bar{\alpha}}\right) \cdot n^3\bar{\alpha}^2 = c_2 \cdot \log^2\left(\frac{8}{n\bar{\alpha}}\right) \cdot n^3\bar{\alpha}^2 \,,$$

(21)

for some constant $c_2 > 0$. Likewise, assuming $n\bar{\alpha} > \frac{1}{2}$, we have from Lemma 1 with probability at least $1 - \frac{1}{n^2\bar{\alpha}} > 0$

$$\left(\sum_{j=1}^{n}\beta_{\sigma(j)}\prod_{i=j+1}^{n}(1-\alpha_{\sigma(i)})\right)^2 \leq c \cdot \log^2\left(8n^2\bar{\alpha}^2\right) \cdot \min\left\{\frac{1}{\bar{\alpha}}, n^3\bar{\alpha}^2\right\} \,.$$

From the law of total expectation, the above, and the fact that the quantity above is deterministically upper bounded by $n^2$ due to the assumptions $\beta_j \leq 1$ and $\alpha_j \leq 1$, we have

$$\mathbb{E}\left[\left(\sum_{j=1}^{n}\beta_{\sigma(j)}\prod_{i=j+1}^{n}(1-\alpha_{\sigma(i)})\right)^2\right] \leq n^2 \cdot \frac{1}{n^2\bar{\alpha}} + c \cdot \log^2\left(8n^2\bar{\alpha}^2\right) \cdot \min\left\{\frac{1}{\bar{\alpha}}, n^3\bar{\alpha}^2\right\} \cdot \left(1 - \frac{1}{n^2\bar{\alpha}}\right)$$

$$\leq \frac{1}{\bar{\alpha}} + c \cdot \log^2\left(8n^2\bar{\alpha}^2\right) \cdot \frac{1}{\bar{\alpha}} = c_3 \cdot \frac{\log^2\left(8n^2\bar{\alpha}^2\right)}{\bar{\alpha}} \,,$$

for some constant $c_3 > 0$. Combining the above with Eq. (21) completes the proof of the proposition. $\qquad\square$

## B.2   Proof of Lemma 1

*Proof.* We will upper bound the expression $\left(\sum_{j=1}^{n}\beta_{\sigma(j)}\prod_{i=j+1}^{n}(1-\alpha_{\sigma(i)})\right)^2$ in two different manners. Taking the minimum of the two will lead to the desired bound.

First, using summation by parts and the fact that $\sum_{j=1}^{n}\beta_{\sigma(j)} = 0$, we have that

$$\left|\sum_{j=1}^{n}\beta_{\sigma(j)}\prod_{i=j+1}^{n}(1-\alpha_{\sigma(i)})\right| = \left|\sum_{j=1}^{n}\beta_{\sigma(j)} - \sum_{j=1}^{n-1}\left(\prod_{i=j+2}^{n}(1-\alpha_{\sigma(i)}) - \prod_{i=j+1}^{n}(1-\alpha_{\sigma(i)})\right)\sum_{i=1}^{j}\beta_{\sigma(i)}\right|$$

$$= \left|\sum_{j=1}^{n-1}\alpha_{\sigma(j+1)}\prod_{i=j+2}^{n}(1-\alpha_{\sigma(i)})\sum_{i=1}^{j}\beta_{\sigma(i)}\right| \leq \sum_{j=1}^{n-1}\alpha_{\sigma(j+1)}\left|\sum_{i=1}^{j}\beta_{\sigma(i)}\right| \,.$$

By the Hoeffding-Serfling bound and a union bound, we have that with probability at least $1 - \delta$, it holds simultaneously for all $j \in \{1, \ldots, n\}$ that $|\sum_{i=1}^{j}\beta_{\sigma(i)}| \leq \sqrt{\log(2n/\delta)j/2} \leq \sqrt{\log(2n/\delta)n/2}$. Plugging into the above, we get that with probability at least $1 - \delta$,

$$\left(\sum_{j=1}^{n}\beta_{\sigma(j)}\prod_{i=j+1}^{n}(1-\alpha_{\sigma(i)})\right)^2 \leq \left(\sum_{j=1}^{n-1}\alpha_{\sigma(j+1)}\sqrt{\frac{\log(2n/\delta)n}{2}}\right)^2$$

$$\leq \left(n\bar{\alpha} \cdot \sqrt{\frac{\log(2n/\delta)n}{2}}\right)^2 = \frac{\log(2n/\delta)}{2} \cdot n^3\bar{\alpha}^2 \,.$$

(22)

We now turn to upper bound the expression $\left(\sum_{j=1}^{n}\beta_{\sigma(j)}\prod_{i=j+1}^{n}(1-\alpha_{\sigma(i)})\right)^2$ in a different manner. To that end, define the index

$$r := \min\left\{n, \left\lceil\frac{6\log(n/\delta)}{\bar{\alpha}}\right\rceil\right\} \in \{1, \ldots, n\} \,.$$

(23)

We first show that $\sum_{j=1}^{n} \beta_{\sigma(j)} \prod_{i=j+1}^{n} (1 - \alpha_{\sigma(i)})$ is close to $\sum_{j=n-r+1}^{n} \beta_{\sigma(j)} \prod_{i=j+1}^{n} (1 - \alpha_{\sigma(i)})$ (namely, where we sum only the last $r$ terms). This is trivially true if $r = n$, so let us focus on the case $r < n$, in which case $r = \left\lceil \frac{6 \log(n/\delta)}{\bar{\alpha}} \right\rceil$. We begin by noting that

$$\prod_{i=1}^{r} (1 - \alpha_{\sigma(i)}) = \exp\left( \sum_{i=1}^{r} \log(1 - \alpha_{\sigma(i)}) \right) \leq \exp\left( -\sum_{i=1}^{r} \alpha_{\sigma(i)} \right).$$

Noting that $\frac{1}{n} \sum_{i=1}^{n} \alpha_i^2 \leq \frac{1}{n} \sum_{i=1}^{n} \alpha_i = \bar{\alpha}$, and using Bernstein's inequality (applied to sampling without replacement, see for example Bardenet et al. [2015, Corollary 3.6]), we have that for any $r \in \{1, \ldots, n\}$, with probability at least $1 - \delta$, it holds that

$$\frac{1}{r} \sum_{i=1}^{r} \alpha_{\sigma(i)} \geq \bar{\alpha} - \sqrt{\frac{2\bar{\alpha} \log(1/\delta)}{r}} - \frac{\log(n/\delta)}{r} \geq \frac{\bar{\alpha}}{2} - \frac{3 \log(1/\delta)}{r},$$

where in the last inequality we used the fact $\sqrt{2xy} \leq \frac{x}{2} + 2y$ for $x, y \geq 0$. Plugging into the previous displayed equation, we have that with probability at least $1 - \delta$,

$$\prod_{i=1}^{r} (1 - \alpha_{\sigma(i)}) \leq \exp\left( -\frac{r\bar{\alpha}}{2} + 3 \log\left( \frac{1}{\delta} \right) \right).$$

Recalling that we assume $r = \left\lceil \frac{6 \log(n/\delta)}{\bar{\alpha}} \right\rceil \geq \frac{6 \log(n/\delta)}{\bar{\alpha}}$ and plugging into the above, it follows that

$$\prod_{i=1}^{r} (1 - \alpha_{\sigma(i)}) \leq \exp(-3 \log(n)) = \frac{1}{n^3}.$$

Since $\sigma$ is a permutation, the same upper bound holds with the same probability for $\prod_{i=n-r+1}^{n} (1 - \alpha_{\sigma(i)})$. Thus, we have that with probability at least $1 - \delta$,

$$
\begin{aligned}
\left| \sum_{j=1}^{n} \beta_{\sigma(j)} \prod_{i=j+1}^{n} (1 - \alpha_{\sigma(i)}) \right| &\leq \left| \sum_{j=n-r+1}^{n} \beta_{\sigma(j)} \prod_{i=j+1}^{n} (1 - \alpha_{\sigma(i)}) \right| + \sum_{j=1}^{n-r} |\beta_{\sigma(j)}| \prod_{i=j+1}^{n} (1 - \alpha_{\sigma(i)}) \\
&\leq \left| \sum_{j=n-r+1}^{n} \beta_{\sigma(j)} \prod_{i=j+1}^{n} (1 - \alpha_{\sigma(i)}) \right| + \sum_{j=1}^{n-r} 1 \cdot \prod_{i=n-r+1}^{n} (1 - \alpha_{\sigma(i)}) \\
&\leq \left| \sum_{j=n-r+1}^{n} \beta_{\sigma(j)} \prod_{i=j+1}^{n} (1 - \alpha_{\sigma(i)}) \right| + \sum_{j=1}^{n-r} 1 \cdot \frac{1}{n^3} \\
&\leq \left| \sum_{j=n-r+1}^{n} \beta_{\sigma(j)} \prod_{i=j+1}^{n} (1 - \alpha_{\sigma(i)}) \right| + \frac{1}{n^2}.
\end{aligned}
\tag{24}
$$

We showed this assuming $r < n$, but the same overall inequality trivially also holds for $r = n$ (with probability 1). Therefore, the inequality holds regardless of the value of $r$ (as defined in Eq. (23)).

To further upper bound this, we note that every term $\beta_{\sigma(j)} \prod_{i=j+1}^{n} (1 - \alpha_{\sigma(i)})$ in the sum above has magnitude at most 1. Applying Azuma's inequality on the martingale difference sequence $\beta_{\sigma(j)} \prod_{i=j+1}^{n} (1 - \alpha_{\sigma(i)}) - \mathbb{E}\left[ \beta_{\sigma(j)} \prod_{i=j+1}^{n} (1 - \alpha_{\sigma(i)}) \big| \sigma(j+1), \ldots, \sigma(n) \right]$ (indexed by $j$ going down from $n$ to $n - r + 1$), we have that with probability at least $1 - \delta$,

$$
\left| \sum_{j=n-r+1}^{n} \left( \beta_{\sigma(j)} \prod_{i=j+1}^{n} (1 - \alpha_{\sigma(i)}) - \mathbb{E}\left[ \beta_{\sigma(j)} \prod_{i=j+1}^{n} (1 - \alpha_{\sigma(i)}) \big| \sigma(j+1), \ldots, \sigma(n) \right] \right) \right|
$$
$$
\leq \sqrt{2r \log\left( \frac{2}{\delta} \right)}.
\tag{25}
$$

Furthermore, since $\sigma$ is a random permutation and $\frac{1}{n}\sum_{i=1}^{n}\beta_i = 0$, the following holds with probability at least $1 - \delta$ simultaneously for all $j \in \{1, \ldots, n\}$, by the Hoeffding-Serfling bound and a union bound:

$$\left| \mathbb{E}\left[ \beta_{\sigma(j)} \prod_{i=j+1}^{n}(1 - \alpha_{\sigma(i)}) \,\Big|\, \sigma(j+1), \ldots, \sigma(n) \right] \right| = \left| \prod_{i=j+1}^{n}(1 - \alpha_{\sigma(i)}) \cdot \frac{1}{j} \sum_{i \in \{1,\ldots,n\}\backslash\{\sigma(j+1),\ldots,\sigma(n)\}} \beta_i \right|$$

$$\leq \left| \frac{1}{j} \sum_{i \in \{1,\ldots,n\}\backslash\{\sigma(j+1),\ldots,\sigma(n)\}} \beta_i \right|$$

$$\leq \sqrt{\frac{\log(2n/\delta)}{2j}} \;.$$

Combining the above together with Eq. (24) and Eq. (25) (using a union bound), we get overall that with probability at least $1 - 3\delta$,

$$\left| \sum_{j=1}^{n} \beta_{\sigma(j)} \prod_{i=j+1}^{n}(1 - \alpha_{\sigma(i)}) \right| \leq \frac{1}{n^2} + \sqrt{2r \log\left(\frac{2}{\delta}\right)} + \sum_{j=n-r+1}^{n} \sqrt{\frac{\log(2n/\delta)}{2j}} \;.$$

Noting[8] that $\sum_{j=n-r+1}^{n} \sqrt{\frac{1}{j}} \leq 2\sqrt{2r}$, plugging into the above and simplifying a bit, we get that with probability at least $1 - 3\delta$,

$$\left| \sum_{j=1}^{n} \beta_{\sigma(j)} \prod_{i=j+1}^{n}(1 - \alpha_{\sigma(i)}) \right| \leq 5\sqrt{r \cdot \log(2n/\delta)} \;.$$

Squaring both sides, plugging in the definition of $r$ and further simplifying a bit, we get that with probability at least $1 - 3\delta$,

$$\left( \sum_{j=1}^{n} \beta_{\sigma(j)} \prod_{i=j+1}^{n}(1 - \alpha_{\sigma(i)}) \right)^2 \leq c' \log^2\left(\frac{2n}{\delta}\right) \cdot \min\left\{ n, \frac{1}{\bar{\alpha}} \right\}$$

for some universal constant $c' > 1$. Combining this with Eq. (22) using a union bound, we have that with probability at least $1 - 4\delta$,

$$\left( \sum_{j=1}^{n} \beta_{\sigma(j)} \prod_{i=j+1}^{n}(1 - \alpha_{\sigma(i)}) \right)^2 \leq c' \cdot \log^2\left(\frac{2n}{\delta}\right) \cdot \min\left\{ n, \frac{1}{\bar{\alpha}}, n^3\bar{\alpha}^2 \right\} \;.$$

Finally, noting that $\min\left\{ n, \frac{1}{\bar{\alpha}}, n^3\bar{\alpha}^2 \right\} = \min\left\{ \frac{1}{\bar{\alpha}}, n^3\bar{\alpha}^2 \right\}$, and letting $\delta' := 4\delta$, we get that with probability at least $1 - \delta'$, the expression above is at most $c_3 \log^2(8n/\delta') \min\left\{ \frac{1}{\bar{\alpha}}, n^3\bar{\alpha}^2 \right\}$ as required. $\square$

### B.3 Remaining Technical Proofs

**Lemma 2.** *Suppose $\sigma_0, \ldots, \sigma_{n-1}$ is a permutation of $\frac{n}{2}$ $0$'s and $\frac{n}{2}$ $1$'s, and that $\eta \leq \frac{1}{\lambda_{\max}n}$. Then*

$$\prod_{i=0}^{n-1}(1 - \eta\lambda_{\max}\sigma_i) \geq 1 - \frac{\eta\lambda_{\max}n}{2} \geq \frac{1}{2} \;.$$

*Proof.* The proof follows immediately from Bernoulli's inequality and the assumption $\eta \leq \frac{1}{\lambda_{\max}n}$. $\square$

---

[8]To see this, note that if $r \geq n/2$, then $\sum_{j=n-r+1}^{n} \sqrt{\frac{1}{j}} \leq \sum_{j=1}^{n} \sqrt{\frac{1}{j}} \leq 2\sqrt{n} \leq 2\sqrt{2r}$, and if $r < n/2$, then $\sum_{j=n-r+1}^{n} \sqrt{\frac{1}{j}} \leq \frac{r}{\sqrt{n-r+1}} \leq \frac{r}{\sqrt{n-n/2+1}} \leq \frac{r}{\sqrt{n/2}} \leq \frac{r}{\sqrt{r}} = \sqrt{r}$.

**Lemma 3.** *Suppose $\sigma_0, \ldots, \sigma_{n-1}$ is a random permutation of $\frac{n}{2}$ 0's and $\frac{n}{2}$ 1's. Then for all $m \in \{1, \ldots, n-1\}$*

$$\mathbb{E}_\sigma \left[ (1 - 2\sigma_0) \prod_{i=1}^{m} \sigma_i \right] = \frac{1}{2} \binom{n/2 - 1}{m - 1} \binom{n - 1}{m}^{-1}.$$

*Proof.* Compute

$$
\begin{aligned}
\mathbb{E} \left[ (1 - 2\sigma_0) \prod_{i=1}^{m} \sigma_i \right] &= \frac{1}{2} \mathbb{E} \left[ \prod_{i=1}^{m} \sigma_i \bigg| \sigma_0 = 0 \right] - \frac{1}{2} \mathbb{E} \left[ \prod_{i=1}^{m} \sigma_i \bigg| \sigma_0 = 1 \right] \\
&= \frac{1}{2} \Pr \left[ \sigma_1 = \ldots = \sigma_m = 1 \big| \sigma_0 = 0 \right] - \frac{1}{2} \Pr \left[ \sigma_1 = \ldots = \sigma_m = 1 \big| \sigma_0 = 1 \right] \\
&= \frac{1}{2} \left( \frac{n/2 \cdot (n/2 - 1) \cdot \ldots \cdot (n/2 - m + 1)}{(n - 1) \cdot (n - 2) \cdot \ldots \cdot (n - m)} \right) \\
&\quad - \frac{1}{2} \left( \frac{(n/2 - 1) \cdot (n/2 - 2) \cdot \ldots \cdot (n/2 - m)}{(n - 1) \cdot (n - 2) \cdot \ldots \cdot (n - m)} \right) \\
&= \frac{1}{2} \left( \frac{(n/2 - 1) \cdot (n/2 - 2) \cdot \ldots \cdot (n/2 - m + 1)}{(n - 1) \cdot (n - 2) \cdot \ldots \cdot (n - m)} \right) m \\
&= \frac{1}{2} \cdot \frac{(n/2 - 1)!}{(n/2 - m)!} \cdot \frac{(n - m - 1)!}{(n - 1)!} m \\
&= \frac{1}{2} \binom{n/2 - 1}{m - 1} \binom{n - 1}{m}^{-1}.
\end{aligned}
$$

$\square$

**Lemma 4.** *Suppose $\sigma_0, \ldots, \sigma_{n-1}$ is a random permutation of $\frac{n}{2}$ 0's and $\frac{n}{2}$ 1's and $\eta \leq \frac{1}{\lambda_{\max} n}$. Then*

$$\mathbb{E} \left[ \sum_{i=0}^{n-1} (1 - 2\sigma_i) \prod_{j=i+1}^{n-1} (1 - \eta \lambda_{\max} \sigma_j) \right] \leq -\frac{1}{8} \eta \lambda_{\max} n.$$

*Proof.* Denote $Y_i := (1 - 2\sigma_i) \left( \prod_{j=i+1}^{n-1} (1 - \eta \lambda_{\max} \sigma_j) \right)$, we expand $Y_i$ to obtain

$$
\begin{aligned}
\mathbb{E} [Y_i] &= \mathbb{E} [1 - 2\sigma_i] + \sum_{m=1}^{n-i} (-\eta \lambda_{\max})^m \mathbb{E} \left[ \sum_{i+1 \leq i_1, \ldots, i_m \leq n-1 \text{ distinct}} (1 - 2\sigma_i) \left( \prod_{l=1}^{m} \sigma_{i_l} \right) \right] \\
&= \sum_{m=1}^{n-i-1} (-\eta \lambda_{\max})^m \binom{n - i - 1}{m} \mathbb{E} \left[ (1 - 2\sigma_0) \prod_{l=1}^{m} \sigma_l \right] \\
&= \frac{1}{2} \sum_{m=1}^{n-i-1} (-\eta \lambda_{\max})^m \binom{n - i - 1}{m} \binom{n/2 - 1}{m - 1} \binom{n - 1}{m}^{-1},
\end{aligned}
$$

where the last equality is by Lemma 3. Denote the $m$-th summand by $a_m$, we bound the quotient of two subsequent terms in the above sum by using the assumption $\eta \leq \frac{1}{\lambda_{\max} n}$, so we get for any $m \geq 1$

$$\left| \frac{a_{m+1}}{a_m} \right| \leq \eta \lambda_{\max} \frac{(n - 2m)(n - m - i - 1)}{2m(n - m - 1)} \leq \frac{1}{2} \eta \lambda_{\max} n \leq \frac{1}{2},$$

thus the above sum which alternates signs and begins with a negative term is upper bounded by

$$\frac{1}{2} (a_1 + a_2) \leq \frac{1}{4} a_1 = -\frac{1}{4} \eta \lambda_{\max} \frac{n - i - 1}{n - 1},$$

where we conclude with

$$\mathbb{E}\left[\sum_{i=0}^{n-1}(1-2\sigma_i)\prod_{j=i+1}^{n-1}(1-\eta\lambda_{\max}\sigma_j)\right] \;=\; \sum_{i=0}^{n-1}\mathbb{E}\left[Y_i\right] \;\leq\; -\frac{1}{4}\eta\lambda_{\max}\sum_{i=0}^{n-1}\frac{n-i-1}{n-1}$$

$$=\; -\frac{1}{8}\eta\lambda_{\max}\frac{n(n-1)}{n-1} \;=\; -\frac{1}{8}\eta\lambda_{\max}n\,.$$

$\square$

## C  Equivalence of Optimization Under Conjugate Transformations

Suppose we are given an orthogonal matrix $O$, an initialization point $\mathbf{x}_0 \in \mathbb{R}^d$ and an optimization problem $F(\mathbf{x}) := \frac{1}{2}\mathbf{x}^\top A\mathbf{x} - \mathbf{b}^\top\mathbf{x} = \frac{1}{n}\sum_{i=1}^n f_i(\mathbf{x})$ where $f_i(\mathbf{x}) = \frac{1}{2}\mathbf{x}^\top A_i\mathbf{x} - \mathbf{b}_i^\top\mathbf{x}$. Define the $O$-conjugate optimization problem as $\tilde{F}(\mathbf{x}) := \frac{1}{2}\mathbf{x}^\top\tilde{A}\mathbf{x} - \tilde{\mathbf{b}}^\top\mathbf{x} = \frac{1}{n}\sum_{i=1}^n \tilde{f}_i(\mathbf{x})$ where $\tilde{f}_i(\mathbf{x}) = \frac{1}{2}\mathbf{x}^\top\tilde{A}_i\mathbf{x} - \tilde{\mathbf{b}}_i^\top\mathbf{x}$, initialized from $\tilde{\mathbf{x}}_0$, whereby $\tilde{A}_i, \tilde{\mathbf{b}}_i, \tilde{\mathbf{x}}_0$ are defined using the following transformations:

$$\tilde{A}_i \;:=\; OAO^\top\,, \quad \tilde{\mathbf{b}}_i \;:=\; O\mathbf{b}_i\,, \quad \tilde{\mathbf{x}}_0 \;:=\; O\mathbf{x}_0\,.$$

In this appendix, we show that the $O$-conjugate optimization problem is equivalent in terms of the sub-optimality rate of without-replacement SGD. More formally, we have the following theorem:

**Theorem 7.** *Suppose we have $F$, $\tilde{F}$ and $O$ as above. Let $\mathbf{x}_t$ and $\tilde{\mathbf{x}}_t$ denote the iterate after performing $t$ steps of without-replacement SGD. Then*

$$\tilde{\mathbf{x}}_t \;=\; O\mathbf{x}_t\,.$$

*And in particular, we have that*

$$\tilde{F}(\tilde{\mathbf{x}}_t) \;=\; F(\mathbf{x}_t)\,.$$

*Moreover, if $F$ satisfies Assumption 2 then so does $\tilde{F}$.*

*Proof.* Using induction, the base case is immediate from the definition of $\tilde{\mathbf{x}}_0$, and we have

$$\tilde{F}(\tilde{\mathbf{x}}_0) \;=\; \frac{1}{2}\mathbf{x}_0^\top O^\top OAO^\top O\mathbf{x}_0 - \mathbf{b}^\top O^\top O\mathbf{x}_0 \;=\; F(\mathbf{x}_0)\,.$$

For the induction step, assume the theorem is true for $t$. We will show it also holds for $t+1$. Compute for all $i \in [n]$

$$\nabla_{\mathbf{x}}\tilde{f}_i(\mathbf{x}) \;=\; \nabla_{\mathbf{x}}\left(\frac{1}{2}\mathbf{x}^\top OA_iO^\top\mathbf{x} - (O\mathbf{b}_i)^\top\mathbf{x}\right) \;=\; OA_iO^\top\mathbf{x} - O\mathbf{b}_i\,.$$

Suppose the next function to be processed in iteration $t+1$ is $f_i$, the update rule of without-replacement SGD therefore satisfies

$$\tilde{\mathbf{x}}_{t+1} \;=\; \tilde{\mathbf{x}}_t - \eta\nabla_{\mathbf{x}}f_i(\tilde{\mathbf{x}}_t) \;=\; O\mathbf{x}_t - \eta OA_iO^\top O\mathbf{x}_t + \eta O\mathbf{b}_i \;=\; O\left(\mathbf{x}_t - \eta(A_i\mathbf{x}_t - \mathbf{b}_i)\right) \;=\; O\mathbf{x}_{t+1}\,.$$

Plugging the above in $\tilde{F}$ we obtain

$$\tilde{F}(\tilde{\mathbf{x}}_{t+1}) \;=\; \frac{1}{2}\mathbf{x}_{t+1}^\top O^\top OAO^\top O\mathbf{x}_{t+1} - \mathbf{b}_i^\top O^\top O\mathbf{x}_{t+1} \;=\; F(\mathbf{x}_{t+1})\,.$$

Lastly, it is readily seen that if $F$ satisfies Assumption 2 then so does $\tilde{F}$ since orthogonal matrices are isometries. $\square$