# OpenReview forum: "Random Shuffling Beats SGD Only After Many Epochs on Ill-Conditioned Problems"
_NeurIPS.cc/2021/Conference — NeurIPS 2021 Spotlight_

### Official Review · Reviewer_WsEL · 2021-07-12

**Rating:** 6
**Confidence:** 3

**Summary:**

This paper compares the efficiency of using different variants of stochastic gradient descent to find the minimum of a function that is the sum of multiple convex functions. It focuses on the case where the minimum of the function is dramatically flatter in some directions than in others. More specifically, it compares using a random sample at each time step, going through the samples in a fixed cycle, and repeatedly going through the entire list of samples in a random order. The conclusion is that the more complicated versions do not significantly outperform regular stochastic gradient descent unless the number of timesteps is unreasonably large.

**Limitations And Societal Impact:**

Yes

**Main Review:**

The paper notes that comparable analyses have been carried out before, but that they treated parameters other than $n$ and $k$ as being constant, and generally gave results less favorable to SGD. This paper's version of the comparison appears to be new, although I am not familiar enough with the literature to be sure.

The paper's organization is good and the proofs are generally solid. However, the functions used for theorem 2 appear to violate assumption 1 due to half of the $f_i$ having no $x_3^2$ term.

This appears to make a useful contribution to the question of how one should try to minimize a convex function chosen from a family by arguing that the more complicated methods of choosing what sample to use when do not necessarily make a useful improvement over straight SGD.

----------------------------edit-------------------------

I stand by my original assessment.

**Time Spent Reviewing:**

8

---

> ### Author Response · Authors · 2021-08-09
> **Review Response**
>
> We thank the reviewer for their positive feedback and time spent reviewing our paper.
>
> Following the reviewer's comment, we will make sure to further polish our paper and make certain that all constructions adhere to their respective assumptions.

---

### Official Review · Reviewer_zgxB · 2021-07-17

**Rating:** 5
**Confidence:** 4

**Summary:**

## I update my score after authors' rebuttal
I thank the authors for their detailed rebuttal. I now understand better why and how the results can be useful. I hope that the authors would do a great job of fixing some of the confusing parts in their work, in particular about the exact novelty of their paper, most notably regarding the conditioning. As promised, I thought more about this work and decided to increase my rating from 4 to 5.
####

This paper presents improved upper and lower bounds for shuffling-based algorithms, which are without-replacement variants of SGD. In particular, the authors consider Random Reshuffling, which reshuffles the functions at each epoch, and Single Shuffle that reshuffles them only once at the very start. For both methods, the authors provide bounds that take into account the conditioning number.

The abstract of the paper about the prior work ignoring the problem geometry looks like a big overstatement, while the title seems to claim something already known from prior literature. Thus, I find the results to have limited motivation, even though both upper and lower bounds are of some interest. Another small flaw of the upper bounds is that they are stated for quadratic objectives.

**Limitations And Societal Impact:**

yes

**Main Review:**

I am very confused by this work. On the one hand, it has new results that contribute to the theoretical development of shuffling-based methods, and it gives new guarantees for both RR and SS. On the other hand, the motivation for these guarantees is unclear. First of all, the works of (Ahn et al.) and (Mishchenko et al.) do provide conditioning numbers in their guarantees. Secondly, the message that RR/SS is only better after some number of epochs is essentially already known. It follows from the existing upper bounds of (Ahn et al.) and (Mishchenko et al.) and existing lower bounds of (Safran and Shamir) and (Rajput et al). All of these bounds show that RR/SS is better eventually but not immediately.

I am also not sure why the authors are so interested in proving upper bounds for quadratic objectives. My understanding is that RR/SS is not used as a solver for quadratic problems but rather is a method widely adopted for training neural networks. From that perspective, it could be much more interesting if the authors studied the nonconvex settings where fewer guarantees are known and little is known about the tightness of existing bounds.

### Small comments
Why not provide the new results in the summary table?

**Time Spent Reviewing:**

4

---

> ### Author Response · Authors · 2021-08-09
> **Review Response**
>
> We thank the reviewer for their comments and feedback.
>
> The works of (Ahn et al.) and (Mishchenko et al.) only provide dependence on the condition number for the upper bounds, and not for the lower bounds. Our analysis reveals that the true dependence on the condition number is somewhat intricate and lies somewhere between the previously known lower and upper bounds, where most notably in our construction it is necessary that the number of epochs k exceeds the condition number for without-replacement to beat with-replacement. Previous upper bounds required that k be in the magnitude of the condition number to provide a non-vacuous guarantee, however to the best of our knowledge our work is the first to show that this requirement is actually necessary to improve upon with-replacement SGD. The lower bounds of (Safran and Shamir) and (Rajput et al.) use constructions where the condition number is constant, and therefore cannot imply meaningful lower bounds in terms of the condition number like the ones we provide in our work.
>
> We agree with the reviewer that the non-convex setting holds interesting challenges and questions, especially from a practical perspective. However, many previous works focused on convex (or even quadratic) functions as cited in our paper. Moreover, in light of our results we argue that even the quadratic case raises important and intricate theoretical questions as to the dependence on the condition number, where quadratic objectives seem like a natural avenue to explore. Indeed, our upper bounds for quadratic objectives contribute by suggesting interesting future directions to study, such as deriving lower bounds that depend on the parameter L and not just on lambda_{\max}. As discussed in the paper, such a result can only be achieved for quadratics if they do not commute, which in our opinion exemplifies how the quadratic case is arguably more intricate and interesting than initially perceived.

---

> > ### Comment · Reviewer_zgxB · 2021-08-09
> > **Is this aspect so valuable?**
> >
> > Thank you for your clarifications. The lower bounds that you present are meaningful and answer a natural question regarding the dependence on the conditioning. However, it doesn't seem to change the overall picture of the comparison between GD, SGD, and RR. We already knew that RR is going to be better only eventually from the conditioning-free lower bounds. The upper bounds guaranteed improvement only after a certain number of passes that depends on the conditioning. Your guarantees state that this is necessary, but in the presence of the other mentioned results, this mostly does not change our understanding of RR. Am I missing anything here? Could you explain to me the value of these results?

---

> > > ### Author Response · Authors · 2021-08-09
> > > **We believe it is valuable due to:**
> > >
> > > a) The condition-free lower bounds suggest that RR *can* be better already when n,k are not too small. For example, in Safran and Shamir, their lower bound for RR (Theorem 1) is smaller than the lower bound for with-replacement already when lambda=1 and n,k are larger than a constant (independent of the condition number which could be arbitrarily large). Here we show that in addition, they *can* be better only when the number of epochs scales strongly with the condition number -- this is a much harsher requirement.
> > >
> > > b) We believe our paper does change our understanding of the convergence of RR, at least in certain cases. For example, when d=1 the condition number is meaningless as our problem is always well-conditioned (see remark 4 in the paper) and we are able to significantly improve upon with-replacement and the previously known upper bounds (i.e. Theorems 4 and 5 in Safran and Shamir which depend on the condition number) without requiring k to scale with the condition number. Moreover, in a less stylized multivariate setting, our bounds also suggest when we can expect without-replacement to have better performance than what is achieved in the worst-case. As discussed in Section 6, this will happen when the spectrum of the Hessians of the individual functions is less 'extreme' and more smoothly transitions between the smallest and largest eigenvalues. This points at important future directions to explore for providing convergence guarantees that do not require k to be in the magnitude of the condition number.

---

> > > > ### Comment · Reviewer_zgxB · 2021-08-10
> > > > **Got your a) but I don't buy your argument b)**
> > > >
> > > > Thanks for the clarification about a). I see your point that when judging an algorithm by lower bounds, the current lower bounds are not sufficient. I don't think that algorithms are judged on lower bounds anyway: the majority of works on new algorithms only compare the upper bounds. I agree that there is some value to lower bounds too. I will later update my review based on our discussion and I may increase the score to acknowledge that this point was explained to me.
> > > >
> > > > I didn't really buy your argument in b). The case d=1 cannot be further away from interesting applications. And the heuristic argument about the Hessian eigenvalues in Section 6 is not a valid result, it seems to be mere speculation, and my understanding is that this is something you'd prefer to look at in future work. So, to me it seems that your arguments in b) are not significant.

---

> > > > > ### Author Response · Authors · 2021-08-10
> > > > > **Thank you for further inquiring into our paper**
> > > > >
> > > > > Perhaps an additional noteworthy comment regarding b) is that our results also drastically improve upon an additional case where we have quadratic terms that commute with \lambda=\lambda_{\max} and L>>\lambda_{\max}. Such problem instances are considered ill-conditioned from the perspective of previous upper bounds, requiring k in the magnitude of the condition number to improve upon with-replacement. However, our results show that this case is in fact well-conditioned since the 'effective' condition number is \lambda_{\max}/\lambda=1, thus we improve upon with-replacement already for constant k,n. This is similar to the 1d case, but it is slightly more general and applies in the multivariate case.
> > > > >
> > > > > Regarding the discussion on conditioning we will add clarifications as the reviewer suggests.

---

> > > > ### Comment · Reviewer_zgxB · 2021-08-10
> > > > **And please make the discussion on conditioning less confusing**
> > > >
> > > > To add to my previous comment, I still find the discussion related to the conditioning to be very confusing, especially in the abstract. In particular, the sentence "However, these works ignore or do not provide tight bounds in terms of the problem’s geometry, including its condition number" needs to be changed. Please make it clear that you only mean the **lower** bounds.

---

### Official Review · Reviewer_erne · 2021-07-21

**Rating:** 8
**Confidence:** 4

**Summary:**

This paper studies lower bounds for without-replacement fixed-step-size SGD  (both single shuffling and random reshuffling) for the finite-sum optimization problem with an attention to dependence on the condition number of the problem. Their primary result is to prove a lower bound on the optimization error (realized by a convex quadratic construction) that suggests that unless the number of epochs $k$ is significantly larger than the condition number, neither single shuffling nor random reshuffling variants of without-replacement SGD outperform with-replacement SGD (previously, upper bounds for the with-replacement variants of SGD were considered to beat without-replacement SGD since often results did not take the condition number into account in the bounds). Moreover, they show their lower bound is tight with respect to a class of quadratics which includes the lower bound instance, suggesting that the analysis is somewhat tight.

The restriction on $k$ being large in order for without-replacement SGD to be more efficient is problematic for practical justification for using without-replacement SGD, since if $k$ is large enough, at some point one will prefer to use methods like conjugate gradient instead of using SGD, thus making the regime in which without-replacement experiences gains irrelevant to practice (since there is a better algorithm).

**Limitations And Societal Impact:**

I didn't see anything that's related to societal impact that needs to be addressed.

The authors were very clear about the limitations of their results,  as well as the opportunities for future work. The limitations of their results are reasonable enough for acceptance.

**Main Review:**

This paper is very well-written and clear, identifies a well-motivated, interesting question and a clean answer. It nicely builds upon previous work in the sub-area, provides improvements on pre-existing analysis, and challenges existing wisdom by highlighting the importance of the condition number parameter. I read the lower bound proofs and they seem correct.  I didn't check the (restricted upper bound) proofs as closely, but in my opinion (though requiring a lot of technical calculations), the restricted upper bounds were not as important to the results. The exposition was also clear.

One thought: It would be nice to say earlier on (somewhere in Section 3) an intuitive reason why the random reshuffling without-replacement lower bound requires three dimensions in the lower bound construction instead of two. (You are using one dimension to get the $1/nk^3 $ dependence, and the other to get the $1/(nk)^2$ dependence, but a little more insight into why the this construction has that dependence fall out would be nice.

**Time Spent Reviewing:**

4

---

> ### Author Response · Authors · 2021-08-09
> **Review Response**
>
> We thank the reviewer for their positive feedback.
>
> The reason our random reshuffling lower bound requires a 3d construction is technical, since it is easier to show the 1/(nk)^2 dependence using a separate construction in an additional dimension. We certainly believe however that with a more involved analysis bounding the expectation of the square of the RHS term in Eq. (5) it would be possible to obtain an identical 2d lower bound. We will add an explanation as proposed.

---

> ### Comment · Reviewer_erne · 2021-08-26
> **Keeping my score the same**
>
> Just noting that I read the responses and other reviews of the authors (as well as their comments, etc.) and will be keeping my score the same.
>
> I think lower bounds are interesting, and there is significant work which goes into showing this lower bound. I buy the argument that this lower bound sheds new light on what was previously known regarding the effect of the condition number on the convergence rate of with-replacement SGD.

---

### Official Review · Reviewer_WHdM · 2021-07-22

**Rating:** 7
**Confidence:** 4

**Summary:**

The paper shows that there exists classes of quadratic objectives where single shuffling or random reshuffling cannot offer improvements over SGD with-replacement sampling unless the number of passes exceeds the condition number of the problem. The paper supplements these lowerbound constructions with matching upperbounds that apply to the class of functions considered in their lowerbound construction. In addition, the paper offers simulations supplementing their theoretical results.

**Ethical Concerns:**

No.

**Limitations And Societal Impact:**

Yes.

**Main Review:**

I found the paper to be an interesting read and the technical results appear correct. The results appear to have little to do with how SGD is run in practice, which potentially explains why the result appears to be at odds with running SGD with and without replacement in practice -- from my experience, SGD with random reshuffling dominates with replacement SGD in terms of its convergence behavior.

One discrepancy can be elaborated as follows: The construction used in this paper is different from the multiplicative noise oracle view of analyzing SGD, where, we do not see a full view of all eigendirections of hessian of the objective when we sample one component of the finite sum. In the construction used by this paper, we get a full view of the hessian based on the construction used say in equation (1) - you could take the second derivative of this function and you get a full rank second derivative matrix. This hardly resembles how SGD is run in practice. See for example, the paper of Bach and Moulines (2013), Defossez and Bach (2015) - which work with least squares with a multiplicative noise oracle. In these cases, we get a rank-1 view of the hessian if we sample one term in a finite-sum.

Moreover, note that when working with the multiplicative noise oracle, the matrix products obtained in SGD recursions do not commute with each other, unless, we deal with specific class of problems where each of the d-dimensions are independent of each other and we can decouple the recursions into a single dimensional recursion. Even in this case, note that we do not receive a full view of the Hessian of the problem, and that reflects more appropriately how SGD is run in practice for finite sum problem instances faced in machine learning settings. Note that in order to get a full view of the hessian, one may require to use a mini-batch size, that in some cases can be as large as the condition number of the problem, in order to see spectral concentration kick in, and for the empirical covariance matrix to start resembling the true Hessian.

1. Non-strongly-convex smooth stochastic approximation with convergence rate O(1/n) - Bach and Moulines (2013)
2. Averaged Least-Mean-Squares: Bias-Variance Trade-offs and Optimal Sampling Distributions - Defossez and Bach (2015)


**Time Spent Reviewing:**

3

---

> ### Author Response · Authors · 2021-08-09
> **Review Response**
>
> We thank the reviewer for the comments.
>
> The current constructions can be modified in a straightforward manner to apply to the setting mentioned by the reviewer, where the quadratic term is rank-1 at each sample (and even more specifically, can be simulated by linear predictors with the squared loss with respect to a certain data distribution). For example, in the construction of Theorem 1, we can have the stochastic function be (\lambda/2)x_1^2 for half of the elements, and (\lambda_max/2)*x_2^2 + the random linear term for the others. In such a case we would get an essentially similar result. We will add a discussion of this. But in any case, we would like to point out that the current construction (where we have a full, noiseless view of the Hessian, rather than a noisy view) only makes our result stronger, since our goal here is to show a negative result (i.e. a lower bound).

---

> > ### Comment · Reviewer_WHdM · 2021-08-26
> > **Re: Author Response**
> >
> > Thanks for the clarification - I missed thinking this through! If the authors could change their statement to be applicable under the multiplicative noise oracle (which I am convinced it does), that'd be great. Thanks!

---

### Decision · Program_Chairs · 2021-09-27

**Decision:**

Accept (Spotlight)

**Comment:**

The writing in this paper is clear. It is well-motivated and informs theory being actively developed in its area. The paper's lower bound accounts for the condition number where previous ones did not, and in doing so significantly strengthens the known conditions under which the community can hope to guarantee, in general, that without-replacement SGD outperforms standard SGD.

Three of four reviewers support accepting this paper. One reviewer's recommendation is just below the acceptance bar, but without a particularly strong argument against acceptance. Some of the criticism from this reviewer is around wording in presentation and motivation. I encourage the authors to consider these as suggestions in their final revisions.

Lower bounds are valuable in a community's pursuit of theory and are technically challenging to establish. Several constructive comments from reviewers have been acknowledged by the authors and I believe will improve the writing and clarity further. Together with sufficiently many positive reviews, I recommend it for acceptance.